



# Zircon micro-inclusions as an obstacle for in situ garnet U-Pb geochronology: An example from the As Sifah eclogite locality, Oman

Jesse B. Walters[1,2,3], Joshua M. Garber[4], Aratz Beranoaguirre[2,3], Leo Millonig[2,3], Axel Gerdes[2,3], Tobias Grützner[2,3], Horst R. Marschall[2,3]

[1]NAWI Graz Geocenter, University of Graz, Graz, 8010, Austria
[2]Department of Geosciences, Goethe-University Frankfurt, Frankfurt am Main, 60438, Germany
[3]Frankfurt Isotope and Element Research Center (FIERCE), Goethe-University Frankfurt, Frankfurt am Main, Germany
[4]Department of Geosciences, The Pennsylvania State University, University Park, PA, USA

*Correspondence to*: Jesse B. Walters (jesse.walters@uni-graz.at)

**Abstract.** Garnet is commonly used to calculate pressure (*P*)-temperature (*T*) histories of metamorphic rocks, as well as to monitor changes in bulk-rock composition (*X*) and deformation (*d*). In situ U-Pb geochronology by laser ablation-inductively coupled mass spectrometry (LA-ICPMS) is a rapid and relatively high spatial resolution technique, which can be used to constrain the timing of the metamorphic *P–T–X–d* histories preserved in garnet. However, the low U contents (low μg/g to ng/g levels) of most common metamorphic garnet crystals presents unique analytical challenges, including potential contamination of the U-Pb system by high-U inclusions, such as zircon, rutile, and monazite. Here we use LA split-stream (SS)-ICPMS analysis to simultaneously measure the U, Th, and Pb isotopes and trace-element contents of eclogite-facies garnet from metamafic rocks at As Sifah, Oman. We observe abundant zircon micro-inclusions (<2 μm) in all five dated samples. Strong linear correlations in U vs Zr contents in the analysed laser-ablation spots plot along garnet–zircon mixing lines, the slopes of which can only be explained by zircon contamination. Despite clear zircon contamination in the trace-element data, the time-resolved laser-ablation U and Pb signals show some irregularities but lack sharp diagnostic spikes typically indicative of inclusions. Instead, zircon micro-inclusions are sufficiently small, abundant, and dispersed over the scale of the laser spot site (193 μm diameter) such that their contribution to the U, Th, and Pb signals is diluted to produce irregular time-resolved signals that have previously not been identified as inclusions.

Analyses affected by contamination result in well-defined U-Pb regression lines that give concordia intercept dates of 94–89 Ma. After screening, only one sample had sufficient inclusion-free analyses and spread in U-Pb ratios to calculate a statistically meaningful date. The calculated concordia intercept date of 71 ± 7 Ma is consistent within uncertainty of previously published garnet–whole rock Sm-Nd peak metamorphic ages. We suggest that the 94–89 Ma ages represent the growth of micro-zircons produced during low-grade metamorphism or hydrothermal alteration of the mafic tuff protolith during the submergence of and sediment deposition on the Arabian margin at this time. To obviate the effect of micro-inclusions in garnet LA-ICPMS U-Pb geochronology, we recommend a careful examination of garnet grains by electron microscopy prior to analysis and determination of background garnet U, Th, Pb contents and Th/U combined with the rejection of analyses with even slight or





moderately irregular signals. We also demonstrate that LASS-ICPMS is a powerful tool to screen for inclusion contamination for in situ U-Pb garnet geochronology, providing confidence in the geologic meaning of the resulting ages.

## 1 Introduction

Garnet is a common rock-forming mineral in igneous and metamorphic rocks. As garnet regularly preserves compositional zoning, overprints or develops static and dynamic microstructures, and participates in numerous cation-exchange and net-transfer reactions, it often used to constrain pressure (*P*), temperature (*T*), chemical (*X*), deformation (*d*) histories of rocks in a large range of tectonic settings (see review in Caddick and Kohn, 2013). Linking these changes in *P–T–X–d* to time is required to constrain the rates and mechanisms of geological processes, as well as their broader context in the

geological timeline. The U(-Th)-Pb decay system has a long-half life, a well-characterized decay constant, and was the first decay system to be used in geochronology; as a result, it is one of the most regularly applied systems in geochronology (see review in Mattinson, 2013). Historical U-Th-Pb geochronology has focused on accessory phases, such as zircon, monazite, xenotime, baddeleyite, and titanite, which contain high contents of radioactive parent isotopes (e.g., U, Th) and high parent/daughter ratios. Dates of these phases are linked to *P–T–X–d* histories through textural and chemical criteria. For

example, depleted rare earth (HREE) element plus Y contents in metamorphic zircon and monazite are often interpreted to result from inter-mineral partitioning with co-crystallising garnet (e.g., Pyle and Spear, 1999; Rubatto, 2002; Gibson et al. 2004). Zircon and monazite are predicted to dissolve during prograde to peak thermal metamorphism through a combination of increasing Zr solubility in major minerals, such as garnet, and dissolution of both zircon and monazite into silicate melts during anatexis (e.g., Pyle and Spear, 2003; Corrie and Kohn, 2011; Kohn et al., 2015; Yakymchuck, 2023). Thus, accessory-

phase geochronology may access only fragments of the record preserved in more complexly zoned minerals, like garnet, that grow over a wider range of *P–T–X–d* conditions.

Direct dating of garnet provides a direct link between isotopic dates and the *P–T–X–d* history recorded by garnet (Baxter et al., 2017). Garnet geochronology was first conducted by van Breemen and Hawkesworth (1980) utilizing the Sm-Nd decay system by isotope dilution-thermal ionization mass spectrometry (ID-TIMS). Both Sm-Nd and Lu-Hf garnet analyses

have also utilized isotope-dilution multi-collector inductively-coupled plasma mass spectrometry (ID-MC-ICPMS) (e.g., Blichert-Toft et al., 1997). Early studies were hampered by contamination of garnet by micro-inclusions, the need for analysis of co-genetic phases to anchor the isochron, sample size limitations, and a long and laborious analytical process (see review in Baxter et al., 2017). Modern Sm-Nd and Lu-Hf garnet geochronology has reduced sample size limitations, allowing for sampling of individual zones by micro-drilling (e.g., Christensen et al., 1989; Pollington and Baxter 2010; Nesheim et al.,

2012) and laser milling (Tual et al., 2022). Additionally, chemical pre-treatment of garnet to dissolve micro-inclusions can largely eliminate the effect of contamination (e.g., DeWolf et al., 1996; Scherer et al., 2000). However, weeks of preparatory work are required for a single isochron date, and sampling of garnet zones is only possible on the scale of mm (laser milling) to cm (microdrilling), resulting in relatively few data compared to accessory phase geochronology. In contrast, *in situ* analysis



by LA-ICPMS offers a rapid alternative to ID-TIMS and ID-MC-ICPMS garnet geochronology with relatively little sample
preparation. Several isotopic systems can be analysed in situ in garnet; some (e.g., Lu-Hf) require in-line separation using
modern collision-cell technology (e.g., Tamblyn et al., 2022; Simpson et al., 2023), whereas U-Pb geochronology can be
performed on a standard single-collector (e.g., Seman et al., 2017; Yang et al., 2018; Millonig et al., 2020; O'Sullivan et al.,
2023) or multi-collector ICPMS instrument (Shu et al., 2024; Bartoli et al., 2024). However, the potential influence of inclusion
contamination on the dates must be thoroughly investigated.

Uranium-lead garnet geochronology was first applied by Mezger et al. (1989), using classic ID-TIMS analysis of the
U-Pb system in almandine-pyrope series regional metamorphic garnet; however, DeWolf et al. (1996) reassessed these data
and concluded that nearly all the measured U ($0.05 – 2$ µg/g) in the garnet aliquots was contributed by inclusions. Indeed, other
authors have explicitly used ID-TIMS dating of U-bearing inclusions in garnet to obtain accessory-phase crystallization dates
(e.g., Lima et al., 2012). In situ U-Pb garnet geochronology by LA-ICP-MS was first conducted by Seman et al. (2017) on
grossular-andradite series skarn garnet, which may contain µg/g levels of U (e.g., DeWolf et al., 1996; Seman et al., 2017;
Wafforn et al., 2018; Burisch et al., 2019; 2023). Due to the analytical difficulties of measuring low U contents in situ, only
recently has the LA-ICPMS approach been applied to almandine-pyrope garnet (e.g., Millonig et al., 2020; Schannor et al.,
2021; Mark et al., 2023; Bartoli et al., 2024; and Shu et al., 2024).

Millonig et al. (2020) conducted U-Pb analyses on regional metamorphic garnet with U contents of <0.1 µg/g by LA-
ICPMS and exhaustively considered the effects of U-rich inclusions on garnet U-Pb data through microbeam sampling of
inclusions and garnet host, grain size analysis of inclusions, and calculating the impact of inclusions on U contents measured
in garnet. They suggested that U-rich inclusions may be identified by examining the time-resolved U signal, Th/U ratios, and
data plotted on the Terra-Wasserburg diagram, where inclusions older or younger than the garnet would fall to the left or right
of the regression line between common and radiogenic Pb, respectively.

However, large diameter (150–200 µg/g) spot sizes are used to achieve sufficient U and Pb signals for LA-ICPMS
analyses, making it difficult to avoid micro-inclusions. Small (i.e., <5 µm) may not be clearly visible in optical microscopy or
when setting spots. If such inclusions are sufficiently abundant and garnet U contents sufficiently low, the U signal may be
theoretically dominated by inclusions without sharp discrete spikes in the time-resolved U signal. Such time-resolved signals
may appear irregular but may not necessarily be distinct from other signal irregularities (e.g., zoning). At such low U contents
in the host garnet, the regression line may be defined entirely by mixing between inclusions and host. If the inclusions are
mostly cogenetic, then the regression line will be narrowly defined and may appear to reflect a garnet date unless more carefully
examined.

Here we analysed ultra-low U garnet (<0.01 µg/g) from the As Sifah eclogite locality, Oman, by laser ablation split
stream (LASS)-ICPMS for simultaneous high-sensitivity analysis of U and Pb isotopes and trace elements. We found that
despite the presence of statistically robust regression lines (MSWD = 0.97–1.93) and reproducible dates (89–94 Ma) among
different samples, these data almost entirely reflect contamination by abundant zircon micro-inclusions (<2 µm diameter).





Incorporation of zircon inclusions elevates measured U contents from <0.01 µg/g in clean garnet to 0.01–2.00 µg/g, resulting in data, which define linear mixing lines in Zr–U space between garnet and zircon. The calculated dates are 10–15 myr older than the 81–77 Ma age of eclogite-facies metamorphism calculated by previous studies using Sm-Nd garnet, U-Pb zircon, U-Pb rutile, and Rb-Sr white-mica geochronology and are inconsistent with existing tectonic models for the timing of the high-*P* stage of subduction metamorphism. These data demonstrate the utility of combining simultaneous analysis of U and Pb isotopes and trace elements with detailed petrographic observations to assess garnet U-Pb geochronology by LA-ICPMS in samples where small or abundant inclusions are suspected. Therefore, the presence of inclusions must be rigorously scrutinized before assigning geological meaning to U-Pb dates of metamorphic garnet and we provide recommendations for the screening of inclusion contamination.

## 2 Geological Background

A suite of high-pressure low-temperature (HP-LT) metasedimentary and metavolcanic rocks crop out within the Saih Hatat tectonic window, which is exposed structurally beneath the obducted Samail Ophiolite in NE Oman (e.g., Goffe et al., 1988; El-Shazly et al., 1990; Searle et al., 1994; 2004; Miller et al., 2002; Hansman et al., 2021). Units within the Saih Hatat window have been correlated with Precambrian to Cretaceous continental margin rocks elsewhere in Oman (e.g., the Jebel Akhdar) (Mann and Hanna, 1990; Searle et al., 2004; Chauvet et al., 2009), and thus represent the most deeply subducted and exhumed portion of the Arabian continental margin (although see Zuccari et al., 2023). The highest-grade rocks in the Saih Hatat occur in two shear-zone bounded exposures: **i)** the Hulw window, which preserves rocks metamorphosed to blueschist-facies conditions, and **ii)** the As Sifah window, which contains structurally lower rocks metamorphosed to eclogite-facies conditions (Fig. 1b). Mafic eclogites (with subsidiary felsic eclogites) occur as boudinaged lenses bounded by metacarbonate and calcschist and are exposed along the shoreline just north of the town of As Sifah. The eclogite facies units in the As Sifah window have been correlated to their stratigraphic equivalents in the Hulw window, which were metamorphosed to lower *P–T* conditions (e.g., Miller et al., 2002). These metavolcanics are interpreted to represent metamorphosed mafic to felsic tuffs emplaced during the late Carboniferous based on ca. 298 and ca. 284 Ma U-Pb zircon protolith ages (Gray et al., 2005; Garber et al., 2021).



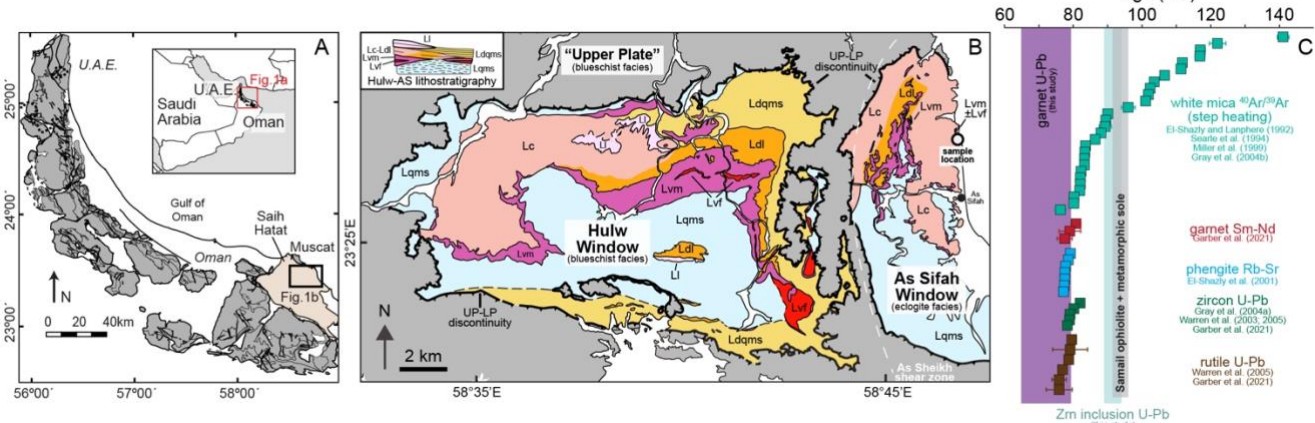

**Figure 1: (a) Regional overview map modified from Nicolas et al. (2000) showing the Samail ophiolite (dark gray) and subducted continental margin rocks of the Saih Hatat (light brown). (b) Lithologic map of the Hulw and As Sifah windows in the Saih Hatat, modified from Miller et al. (2002), Searle et al. (2004), and Warren and Miller (2007) (and lightly modified from Garber et al., 2021).**

**The dominant structural feature is the "UP-LP" discontinuity (bold line), which separates "Upper Plate" rocks (dark gray) from "Lower Plate" units. The poorly-exposed but structurally necessary "As Sheikh" ductile shear zone between the Hulw and As Sifah Windows is shown in a dashed white line. Unit abbreviations: Lqms = lower-plate quartz-mica schist; Lvf = lower-plate felsic volcanic, Lvm = lower-plate mafic volcanic; Lc = lower-plate calcschist and quartz schist; Ldl = brown dolomite; Ldqms = dolomitic quartz-mica schist; Ll = Permian metacarbonate. The mafic eclogite samples in this study are interpreted to be part of the "Lvm"**

**and "Lvf" units exposed just north of the town of As Sifah. The inset stratigraphic sketch is modified from Miller et al. (2002); the Hulw and As Sifah Windows are interpreted as stratigraphically correlative equivalents of the Ordovician (Lqms; Amdeh Fm.) to Permian (Lc-Ll; Saiq Fm.) Arabian continental margin sequence that is also exposed in the "Upper Plate" (e.g., Searle et al., 2004; Warren and Miller, 2007; Chauvet et al., 2009). (c) Previously published geochronology from the As Sifah eclogites, including the new U-Pb dates in this study. Ar/Ar hornblende and U-Pb zircon geochronology from the overlying Samail ophiolite and**

**metamorphic sole is shown as a gray bar (dates from Hacker et al., 1996; Warren et al., 2005; Styles et al., 2006; Rioux et al., 2012; 2013; 2016; 2021; 2023). Where not visible, error bars on dates are smaller than symbols. See text for additional discussion.**

Previous studies have constrained peak eclogite-facies *P–T* conditions in the As Sifah eclogites to 2.0–2.5 GPa at ~550 °C through a combination of garnet–clinopyroxene–phengite thermobarometry, Si-in-phengite barometry, and equilibrium thermodynamic modelling (Searle et al., 1994; Warren and Waters, 2006; Massonne et al., 2013), whereas El-

Shazly (2001) suggested significantly lower pressures (<1.6 GPa). The As Sifah eclogites are thought to have undergone metamorphism along a single clockwise *P–T* path along a "cold" subduction geothermal gradient (El-Shazly et al., 1990; Warren and Waters, 2006; Garber et al., 2021), as evidenced by pseudomorphs after lawsonite and inclusions of chloritoid preserved in garnet from some lithologies (e.g., Fig 2). Eclogite-facies assemblages were partially hydrated at high *P*, with retrograde phengite and amphibole overprinting peak garnet and omphacite, and variably overprinted at greenschist facies

conditions by the assemblage epidote + albite + chlorite + calcic amphibole + hematite + carbonate + titanite (e.g., El-Shazly et al., 1990; Massonne et al., 2013; Garber et al., 2021).

Extensive geochronological studies have been conducted at As Sifah (Fig. 1c), leading to myriad hypotheses for the age and tectonic configurations leading to HP-LT metamorphism. Uranium-lead zircon and rutile geochronology on mafic eclogites conducted by TIMS resulted in dates of 79–78 Ma (Warren et al., 2003; Warren et al., 2005), which was suggested

to represent the timing of a single eclogite-facies metamorphic event based on abundant zircon and rutile inclusions in both



garnet and omphacite as well as in the matrix. Gray et al. (2004a) subsequently conducted Sm-Nd garnet–whole rock isochron geochronology, and obtained dates of 110 ±9 and 109 ±13 Ma, which were interpreted to represent the timing of peak eclogite-facies metamorphism, with the <80 Ma U-Pb zircon and rutile dates interpreted as relating to late exhumation. These older dates were nominally supported by numerous K-Ar and $^{40}$Ar/$^{39}$Ar step-heating white mica dates spanning 140–80 Ma that were

reported by several studies (Montigny et al., 1988; El-Shazly and Lanphere, 1992; Searle et al., 1995; Miller et al., 1999; Gray et al., 2004b), leading some authors to infer distinct HP-LT metamorphic events at 130–110 Ma and 80 Ma. However, El-Shazly et al. (2001) measured several ca. 80 Ma Rb-Sr isochron dates from As Sifah micas, and suggested that >80 Ma K-Ar and $^{40}$Ar/$^{39}$Ar dates were the result of inherited non-radiogenic excess Ar (Kelley, 2002). This latter interpretation is supported by the direct observation of extensive excess Ar by high-precision laser $^{40}$Ar/$^{39}$Ar dating of individual micas from As Sifah

(Warren et al., 2011), which is consistent with the derivation of excess Ar from trapped pore fluids during devolatilization (Smye et al., 2013). Recently, Garber et al. (2021) determined new Sm-Nd garnet–whole rock ID-TIMS dates from multiple As Sifah lithologies, and obtained 81–77 Ma dates with internally consistent εNd intercepts, as well as providing additional U-Pb zircon (80–78 Ma) and rutile (80–76 Ma) dates that are consistent with previous U-Pb geochronology. A comparison between the Garber et al. (2021) and Gray et al. (2004a) Sm-Nd data showed that the earlier determined ~110 Ma dates most

likely represent mixing lines rather than isochrons. These recent studies demonstrate that previously published K-Ar, $^{40}$Ar/$^{39}$Ar, and Sm-Nd dates from As Sifah are spurious, and that the As Sifah rocks achieved peak-metamorphic conditions only after ~81 Ma.

Along with structural data, these disparate dates from different studies have led to greatly contrasting tectonic interpretations. The Samail ophiolite, a relatively intact segment of obducted upper oceanic lithosphere, is thought to have

initially crystallized between 96.2 and 95.7 Ma based on high-precision ID-TIMS zircon U-Pb geochronology (Rioux et al., 2012; 2013; 2021). The metamorphic sole welded to the base of the ophiolite – thought to have formed during the initial stage of Samail ophiolite obduction – consists of metasediments and metabasalts metamorphosed to granulite facies conditions of 700–900 °C and 0.8–1.4 GPa (Ghent and Stout, 1981; Gnos, 1998; Hacker and Mosenfelder, 1996; Searle and Cox, 2002; Cowan et al., 2014; Soret et al., 2017; Ambrose et al., 2021). Though there is disagreement based on recently published,

disparate Lu-Hf garnet dates (e.g., Guilmette et al., 2018), the timing of high-grade metamorphism as dated by high-precision U-Pb zircon and hornblende $^{40}$Ar/$^{39}$Ar geochronology is coeval with or at most <500 kyr prior to the formation of the ophiolite crust (Hacker et al., 1996; Warren et al., 2005, Rioux et al., 2016; 2023). The spatial and temporal correlation of ophiolite and sole formation has been used to construct a tectonic model in which subduction initiation occurred simultaneously with the formation of the ophiolite crust in a suprasubduction zone setting (Searle and Cox, 2002; Cowan et al., 2014; Soret et al., 2017;

Kotowski et al., 2021), consistent with the magmatic compositional evolution of the ophiolite itself (e.g., Pearce et al., 1981; Ishikawa et al., 2002; Kusano et al., 2017; Belgrano and Diamond, 2019; Rioux et al., 2021). This interpretation, however, results in strongly contrasting thermal regimes between the granulite facies rocks of the metamorphic sole and far cooler blueschist and lawsonite eclogites of the Saih Hatat, which were only separated in time by ~15 Myr. The simplest kinematic explanation is a single subduction system, starting with high-*T* conditions during subduction initiation below the ophiolite



crust at 96–95 Ma, followed by relaxing of the geothermal gradient for ~10 million years until subduction of the Arabian continental margin at lawsonite-eclogite-facies conditions at ca. 80 Ma, coincident with the final emplacement of the Samail ophiolite (e.g., El-Shazly et al., 2001; Searle et al., 1994; Warren et al., 2003; Agard et al., 2010; Duretz et al., 2016; Garber et al., 2021). This interpretation is further consistent with biostratigraphic constraints for the initial timing of the drowning of the Arabian passive margin, at ~94 Ma (Robertson, 1987). In contrast, studies which support an older age or protracted HP-

LT metamorphism have invoked multiple subduction zones operating simultaneously to account for the vastly different thermal regimes between the HT metamorphic sole rocks and the LT Saih Hatat rocks (e.g., Gray et al., 2004a; 2004b; Goscombe et al., 2020; Ring et al., 2024). Additional geochronological constraints may help differentiate a geologically accurate tectonic model for HP-LT metamorphism and ophiolite obduction and was the motivation behind this study.

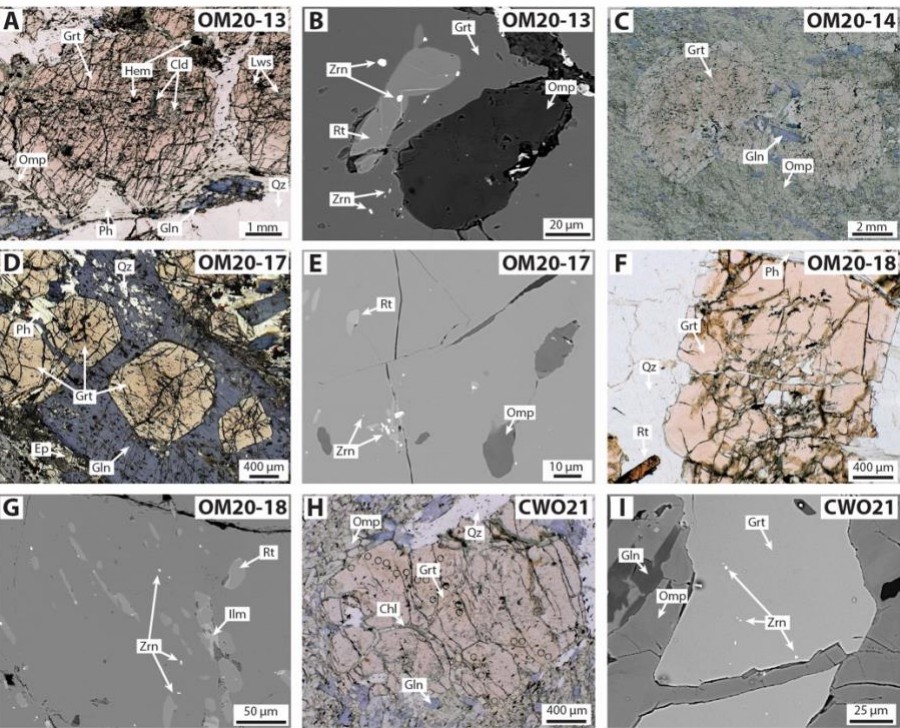

**Figure 2: Transmitted optical light (a, b, d, f, and h) and backscattered electron (b, e, g, and i) images of the studied samples. (a, b) Garnet porphyroblast with inclusions of omphacite, chloritoid, Ti-hematite, rutile, zircon, and pseudomorphs after lawsonite in a quartz and glaucophane-rich banded eclogite (OM20-13). (c) Garnet porphyroblasts in an omphacite-rich eclogite with minor retrograde glaucophane (OM20-14). (d) Retrogressed eclogite with large mm-scale glaucophane grains overgrowing peak metamorphic garnet with phengite and epidote in the matrix (OM20-17). (e) Omphacite, rutile, and zircon inclusions in garnet from**
**OM20-17. (f, g) Garnet porphyroblast with inclusions of quartz, rutile, ilmenite, and zircon in a quartz–garnet segregation with minor phengite (OM20-18). (h) Garnet porphyroblast with omphacite, quartz, and retrograde glaucophane and chlorite in eclogite CWO21. (i) Backscattered electron image with micro-zircon inclusions in garnet in sample CWO21 from Warren (2004). Mineral abbreviations after Warr (2021).**



## 3 Sample Selection

Five samples of mafic to felsic eclogite with varying mineralogical composition were selected for analysis. Four were collected from Oman in March 2020 (OM20-13, OM20-14, OM20-17, and OM20-18), whereas the fifth (CWO21) was previously characterized and dated by Warren et al (2003; 2005) and Garber et al. (2021). The samples all exhibit the peak metamorphic assemblage of garnet + omphacite + rutile ± quartz, as described by previous authors (e.g., Fig. 2), but with varying proportions of these phases: Samples OM20-14 and CWO21 have high proportions of omphacite (Fig. 2c and h)

whereas sample OM20-18 is a felsic high-$P$ rock with garnet + quartz with minor phengite and no omphacite (Fig. 2f). Garnet grains in sample OM20-13 contain an inclusion assemblage of chloritoid, clinozoisite + paragonite after lawsonite, and titanohematite (Fig. 2a), whereas garnet in other samples includes only the peak metamorphic assemblage. Samples are variably retrogressed to both the blueschist- (especially phengite + glaucophane–riebeckite) and greenschist-facies assemblages. In sample OM20-13, omphacite is rarely observed and oriented phengite and glaucophane overgrow the peak

assemblage (Fig. 2a). In sample OM20-17, garnet grains are completely encased in cm-scale blades of glaucophane, and matrix omphacite is largely replaced by a fine-grained symplectite (Fig. 2d). Retrograde glaucophane is observed in OM20-14 and, to a lesser extent, in CWO21. Garnet grains are highly fractured in all samples, with chlorite + hematite occurring within some fractures.

Accessory phases are abundant in all samples. The inclusions in garnet in OM20-13 evolve from titanohematite (with

ilmenite exsolution) in the core to rutile in the rim, suggesting a prograde partitioning of $Fe^{3+}$ into aegirine and Ti into rutile. The other samples only contain rutile or rutile + ilmenite (among the Ti minerals) as inclusions in garnet and in the matrix. A second generation of hematite with square-shaped to bladed habits are observed in the matrix and in fractures in garnet, associated with carbonate, chlorite, and fine-grained symplectites. Frequent zircon micro-inclusions (<1–2 µm diameter) are visually observed primarily as inclusions in garnet in all five samples (Fig. 2), where sample OM20-18 was qualitatively

observed to have the fewest inclusions overall.

## 4 Analytical methodologies

### 4.1 Sample preparation

Four samples (OM20- samples) were prepared as ~100 µm thick polished sections, whereas CWO21 was polished to the standard thickness (30 µm). The sections were examined petrographically and imaged by optical microscopy using a

petrographic microscope, as well as a Keyence VHX-6000 digital microscope for full section transmitted light scans.

### 4.2 Electron Probe Microanalysis

Electron probe microanalysis (EPMA) was conducted at GUF using a JXA-8530F Plus Hyperprobe field-emission EPMA equipped with five WDS carrying standard-type as well as H- and L-type spectrometer crystals. Prior to analysis,



backscattered electron imaging and phase confirmation by energy-dispersive spectroscopy was conducted. Operating

conditions for quantitative WDS analysis were a 15 kV accelerating voltage, 20 nA beam current, and a 3 μm spot size. Elements analysed were Si, Ti, Al, Cr, Fe, Mn, Mg, Ni, Ca, Na, K, and P. On-peak counting times of 20 s (10 s background) were used for all elements. Natural and synthetic reference materials used for calibration were albite (Na), forsterite (Mg), fayalite (Fe), wollastonite (Ca, Si), pyrophanite (Mn), $Cr_2O_3$ (Cr), $Al_2O_3$ (Al), and $KTiOPO_4$ (Ti, K, P). Garnet structural formulae were calculated using MinPlotX (Walters, 2022; Walters and Gies, 2024). All EPMA data are reported in Table S1

and garnet transects are plotted in Figure S1. Garnet grains from samples OM20-14 and CWO21 were not analysed by EPMA.

### 4.3 LASS-ICP-MS garnet analysis

Laser ablation split-stream inductively-coupled plasma mass spectrometry (LASS-ICPMS) analyses were performed during two sessions at the Frankfurt Isotope and Element Research Center (FIERCE), GUF. Ablation was performed using a RESOLUTION-LR (Resonetics)193-nm ArF Excimer laser (Compex Pro 102, Coherent) equipped with a two-volume (Laurin

Technic S155) ablation cell. A spot size of 193 μm diameter (round) was used for all analyses. A fluence of ~2 $J/cm^2$ at 12 Hz was used, with a resulting ablation rate of ~0.6 μm/s. Acquisition of 18 s of background and 18 s of ablation was conducted following 4 pulses of pre-ablation. Prior to LASS-ICPMS analysis, transmitted light optical microscopy and BSE imaging were used to preselect locations of apparently inclusion free garnet domains.

Simultaneous analysis of U-Pb isotopic ratios and trace element contents were measured by splitting the ablation

stream between two mass spectrometers following the protocol of Shu et al. (2024). Approximately 80 % of the volume was sent to a ThermoScientific Neptune Plus sector field multi-collector ICPMS for U-Pb isotopic analysis, whereas the remaining 20 % was diverted to a ThermoScientific Element-XR sector-field ICPMS for trace-element analysis. For U-Pb analysis, $^{206}Pb$, $^{207}Pb$, and $^{208}Pb$ were measured using secondary-electron multipliers, whereas $^{232}Th$ and $^{238}U$ were measured on Faraday cups equipped with $10^{13}$ Ω amplifiers. Signal strength was tuned for maximum sensitivity while maintaining an oxide formation of

~0.5 % UO/U and low element fractionation (Th/U of ~0.9 for NIST-SRM614). The calibration strategy for U-Pb analyses includes the soda-lime glass NIST-SRM614 (Jochum et al., 2011) as the primary reference material to correct for instrumental drift, mass bias, and inter-element fractionation. A Mali garnet reference material (202.0 ±1.2 Ma, TIMS; Seman et al., 2017) was used to correct for the matrix offset between NIST glass and garnet. Secondary reference materials Lake Jaco Pink (34.0 ±1.4 Ma, LA-ICPMS; Seman et al., 2017) and an in-house garnet reference material (Pakistan3b; ca. 45 Ma) were also analysed

to test for reproducibility. Additional analytical details are given in supplementary Table S2A.

Raw data were processed and corrected offline using the in-house VBA spreadsheet program of Gerdes & Zeh (2006, 2009), which employs the algorithms of Ludwig (2012). A minor drift correction using NIST-SRM614 was also applied over each session using 3rd ($^{207}Pb/^{206}Pb$, $^{232}Th/^{238}U$) and 4th ($^{206}Pb/^{238}U$) order polynomial functions. Analyses of Mali garnet were used to calculate offset factors applied to $^{206}Pb/^{238}U$ ratios, accounting for differences in ablation rates and composition between

NIST-SRM614 and garnet. Calculated offset factors are 0.91 and 0.96 for Sessions 1 and 2, respectively. Data were used to



define linear arrays in $^{207}Pb/^{206}Pb$ vs. $^{238}U/^{206}Pb$ space (Tera and Wasserburg, 1972) from which pooled lower-intercept concordia ages were calculated. The initial $^{207}Pb/^{206}Pb$, or 'common Pb', ratios are calculated as the y-intercept of a linear regression of the data array and are not anchored for unknown samples, Lake Jaco Pink, and Pakistan3b. Following Horstwood et al. (2016), quoted age uncertainties ($2\sigma$ absolute) include within-run precision, counting statistics, background, and excess

of scatter (calculated from the primary RM). An excess of variance is calculated from the offset reference material and added quadratically (see Table S2). Systematic uncertainties are reported as an expanded uncertainty, which considers ratio uncertainty (1 %, $2\sigma$), long-term reproducibility (1.5 %, $2\sigma$), and decay-constant uncertainties. Measured dates of Lake Jaco Pink (36.6 ±1.4 and 35.7 ±1.4 Ma, 2s) are within uncertainty of the LA-ICPMS age of 34.0 ±1.4 Ma by Seman et al. (2017). Additionally, the 45.4 ±0.8 Ma (2s, both sessions) date of Pakistan3b are consistent with our long-term reproducibility for this

secondary reference material. Tera Wasserburg diagrams for secondary reference materials are shown in Figure S2. These data suggest that dates are both precise and true relative to reference values and support high long-term reproducibility. The ablation rate of garnet is ~50 % lower compared to the ablation rate of glass. To account for this effect, we apply an average pit-depth correction measured on garnets during previous sessions (see Millonig et al., 2020). Calculated total Pb considers both the radiogenic and common Pb.

Trace elements were measured simultaneously on the Element-XR ICPMS. Masses $^{23}Na$, $^{25}Mg$, $^{27}Al$, $^{29}Si$, $^{43}Ca$, $^{44}Ca$, $^{45}Sc$, $^{49}Ti$, $^{51}V$, $^{53}Cr$, $^{55}Mn$, $^{57}Fe$, $^{59}Co$, $^{66}Zn$, $^{85}Rb$, $^{88}Sr$, $^{89}Y$, $^{90}Zr$, $^{93}Nb$, $^{133}Cs$, $^{137}Ba$, $^{139}La$, $^{140}Ce$, $^{141}Pr$, $^{146}Nd$, $^{147}Sm$, $^{151}Eu$, $^{157}Gd$, $^{159}Tb$, $^{161}Dy$, $^{165}Ho$, $^{167}Er$, $^{169}Tm$, $^{172}Yb$, $^{175}Lu$, $^{178}Hf$, $^{181}Ta$, $^{208}Pb$, $^{232}Th$, and $^{238}U$ were analysed by peak jumping in pulse counting mode with a total time per cycle of 0.51 ms for 54 cycles (total acquisition time of 41.6 s). Data were processed using LADR (Norris & Danyushevsky, 2018) using $^{29}Si$ as the internal reference isotope with Si measured by EPMA for

OM20-13, OM20-17, and OM20-18. Garnet grains in OM20-14 and CWO21 were not analysed by EPMA, and the Si content of garnet measured by EPMA for OM20-13 was used for these analyses (see Table S3). Analysed reference materials include basalt glass BIR-1G (Jochum et al., 2005) and NIST-SRM614 (Jochum et al., 2011). BIR-1G has extremely low U and Th contents (0.023 and 0.030 µg/g, respectively; see Table S3); therefore, we use NIST-SRM614 as the primary reference material and BIR-1G as the secondary reference for all masses with some exceptions. We found variable and/or low signals of masses

$^{25}Mg$, $^{45}Sc$, $^{49}Ti$, $^{53}Cr$, $^{55}Mn$, $^{57}Fe$, $^{59}Co$, $^{66}Zn$, and $^{90}Zr$ for analyses of NIST-SRM614. For these elements we instead use BIR-1G as the primary reference material and NIST-SRM614 as the secondary reference material (see Table S3). Uncertainties are quoted as 2s absolute and include background signal, peak signal, and reference material analysis uncertainty, as well as propagated systematic uncertainties in the published reference material values. Trace-element data are summarized in Table S3. All trace-element contents for NIST-SRM614 are within uncertainty of the GeoREM preferred values, except for U, which

is ~80 % higher (1.51 µg/g vs. 0.823 µg/g) in Session 1. Note that total Pb is calculated from the $^{208}Pb$ signal assuming all Pb is common Pb with a present day Pb composition.



## 4.4 LA-ICPMS garnet trace element analysis

After the split-stream analysis, new polished thick sections were prepared and garnet grains in samples OM20-13, OM20-17, and OM20-18 were reanalysed for their trace elements by LA-ICPMS on the Element-XR using a 33 μm diameter

spot size to avoid micro-inclusions. Ablation was performed in a He atmosphere (0.3 l/min) and mixed in the ablation funnel with 1.0 l/min Ar and 0.05 l/min $N_2$. A fluence of ~4 J/cm$^2$ at a repetition rate of 6 Hz was used. Acquisition of 20 s of background and 20 s of ablation followed 4 pulses of pre-ablation. Further ablation analytical details are given in supplementary Table S4. Masses $^{23}$Na, $^{25}$Mg, $^{27}$Al, $^{29}$Si, $^{43}$Ca, $^{44}$Ca, $^{49}$Ti, $^{51}$V, $^{52}$Cr, $^{55}$Mn, $^{57}$Fe, $^{59}$Co, $^{60}$Ni, $^{66}$Zn, $^{88}$Sr, $^{89}$Y, $^{90}$Zr, $^{93}$Nb, $^{139}$La, $^{140}$Ce, $^{141}$Pr, $^{146}$Nd, $^{147}$Sm, $^{151}$Eu, $^{157}$Gd, $^{158}$Gd, $^{159}$Tb, $^{161}$Dy, $^{165}$Ho, $^{167}$Er, $^{169}$Tm, $^{172}$Yb, $^{175}$Lu, $^{178}$Hf, $^{181}$Ta, $^{206}$Pb,

$^{208}$Pb, $^{232}$Th, and $^{238}$U were analysed by peak jumping in pulse counting mode with a total time per cycle of 1.007 s for 44 cycles (total acquisition time of 44.3 s). Basalt glass GSD-1G (Jochum et al., 2005) was used as the primary reference material and basalt glass BIR-1G, basalt glass BCR-1, and soda lime glass NIST-SRM612 (Pearce et al., 1997; Jochum et al., 2005; 2011) were used as the secondary reference materials. Trace-element data processing and uncertainty reporting are the same as above. Measured elemental contents are within uncertainty to the GeoREM preferred values (see Table S4). Total Pb is

calculated the same as above.

## 4.5 LA-ICPMS garnet trace element mapping

Laser-ablation inductively coupled plasma mass spectrometry (LA-ICPMS) for trace elements was performed on garnet in sample CWO21 at the LionChron facility at The Pennsylvania State University, USA. Samples were ablated using a Teledyne/Photon Machines Analyte G2 excimer laser ablation system with a Helex2 ablation cell, coupled to a Thermo

Scientific iCAP-RQ ICPMS system for trace elements. The total Ar gas flow for the experiment was 1.01 L/min, with total He gas flows from the laser at 0.44 L/min. The CWO21 garnet was run as a grid map in a single session, with a 20 μm square, 15 Hz repetition rate, 120 shots, and a laser fluence at the sample surface of ~4.5 J/cm$^2$ for each pixel. The laser was first fired thrice for each pixel location to remove surface contamination; only short, 1–2 s pauses were run between each pixel, and reference material analyses and backgrounds were collected once every ~200 unknowns. Analyses of unknowns were

bracketed by analyses of whole-rock glasses Kilauea basalt KL2-G and Alpine quartz diorite T1G from the Max-Planck-Institut (Jochum et al., 2006), as well as NIST-SRM612 glass (Pearce et al., 1997; Jochum et al., 2011). KL2-G was used as the primary reference material for all analyses. For trace-element quantification, $^{27}$Al (assuming 11.0 wt. % Al) was used as the internal reference isotope, with measured peaks on the iCAP-RQ at $^7$Li, $^{23}$Na, $^{24}$Mg, $^{27}$Al, $^{29}$Si, $^{31}$P, $^{43}$Ca, $^{45}$Sc, $^{49}$Ti, $^{51}$V, $^{52}$Cr, $^{55}$Mn, $^{57}$Fe, $^{88}$Sr, $^{89}$Y, $^{90}$Zr, $^{146}$Nd, $^{147}$Sm, $^{153}$Eu, $^{157}$Gd, $^{159}$Tb, $^{163}$Dy, $^{165}$Ho, $^{166}$Er, $^{169}$Tm, $^{172}$Yb, $^{175}$Lu, $^{178}$Hf, $^{208}$Pb, and $^{238}$U.

Iolite version 4 (Paton et al., 2011) was used to correct measured isotopic ratios and elemental intensities for baselines, plasma-induced fractionation, and instrumental drift. The mean and standard error of the measured ratios of the backgrounds and peaks were calculated after rejection of outliers more than two standard errors beyond the mean. Using the same methods as applied





to unknowns, this routine yielded values for T1G accurate to within 5 % for all elements except for Pb (10 %) and U (15 %). LA-ICPMS maps for all elements are given in Figure S3.

## 4 Results

### 4.1 Characterization of garnet compositional zoning and textures

Garnets in samples OM20-13, OM20-17, and CWO21 show bell-shaped major-element compositional patterns (Figs. 3a, S1, and S2), comparable to garnet compositional profiles reported in previous studies of blueschist and eclogite at As Sifah (Warren & Waters, 2006; Garber et al., 2021). In the EPMA transect of OM20-13, the almandine content decreases ($X_{Alm}$ = 0.76 to 0.71) and pyrope content increases ($X_{Prp}$ = 0.08 to 0.20) from core to rim (Fig. S1a). In contrast, in OM20-17 the pyrope content also increases from core to rim ($X_{Prp}$ = 0.04 to 0.22), but almandine increases from $X_{Alm}$ = 0.53 in the core to 0.71 in the mantle and then decreases to 0.67 in the rim (Fig. S1b). Elemental LA-ICPMS maps of sample CWO21 similarly show a decrease in Fe and increase in Mg from garnet core to rim (Fig. S2). In contrast, garnet in OM20-18 displays oscillatory zoning in Fe, Mg, and Mn, where high Mg + Fe zones are balanced by low Mn. In all four samples, there is a general core to rim decrease in garnet Mn content (Figs. 3a and S1).

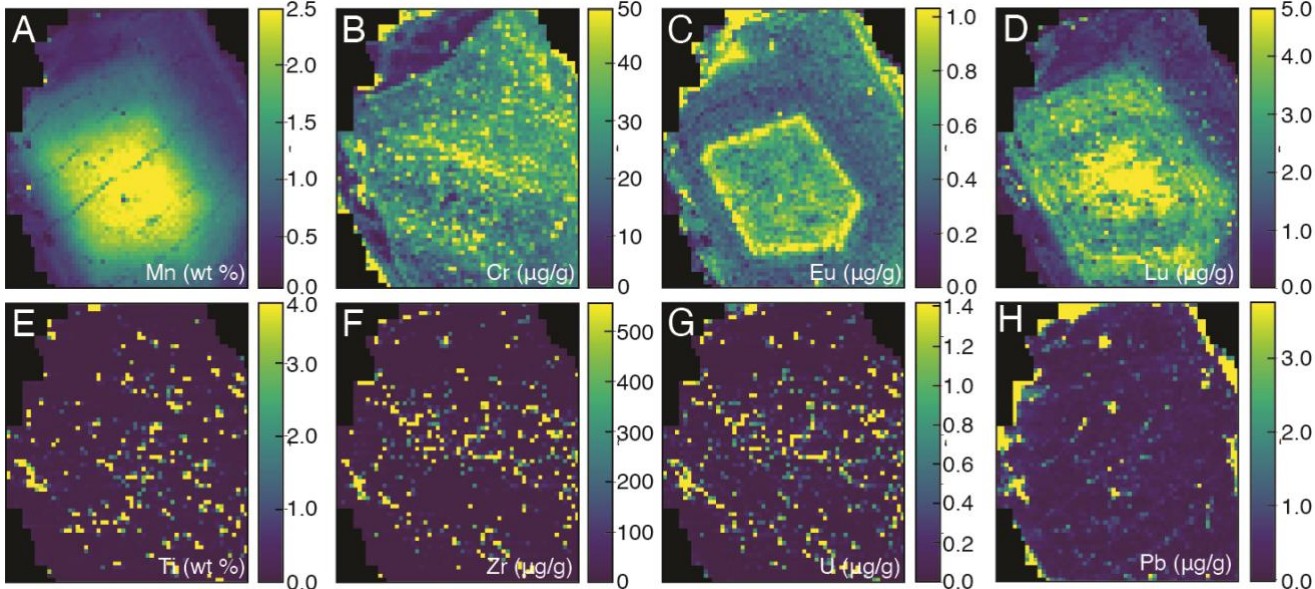

**Figure 3: LA-ICMS minor- and trace-element maps of Mn (a), Cr (b), Eu (c), Lu (d), Ti (e), Zr (f), U (g), and Pb (h) in a garnet grain in sample CWO21. Each pixel is 20 x 20 μm².**

Trace element maps of CWO21 reveals complex zoning patterns. For example, Cr contents are relatively uniform throughout the mapped garnet (30–40 μg/g), except for a portion of the rim in which the Cr content drops to <10 μg/g (Fig. 3b). Middle REE, such as Eu, show relatively high contents in the core surrounded by a high-concentration annulus and lower





contents in the mantle and rim (Fig. 3c). Complex oscillatory zoning is observed in Y + HREE, with uniformly high contents in the core followed by an oscillatory-zoned inner mantle and low Y + HREE contents in the outer mantle and rim (Fig. 3d). Maps of Ti, Zr, U and Pb show that these elements are elevated only in inclusion phases, whereas uniformly low contents are observed in the background garnet except for Ti (Fig. 3e–h).

## 4.2 Concentrations of Zr, Ti, Nb, Ce, Th, U and Pb in garnet

LA-ICPMS trace-element analyses with a small size of the laser spot (33 μm diameter) were collected to minimize potential inclusion contamination and measure 'clean' garnet signals. These analyses were conducted separately, i.e., after the U-Pb analyses had been performed. Micro-inclusions of zircon, as well as abundant inclusions of rutile, Ti-hematite, and ilmenite were observed petrographically and by EPMA analysis and by LA-ICPMS mapping (Figs. 2-3). While analysis locations were pre-selected using optical and BSE imaging, the large volume of ablated material using the 193 μm diameter spot for U-Pb analysis requires careful consideration of potential inclusion contamination. Here, we assess the minor- and trace-element results from the 33-μm-diameter spot analyses to establish the background composition of uncontaminated garnet. An emphasis is placed on elements, which exhibit high contents in potential contaminant phases, such as Zr in zircon, Ti and Nb in rutile, ilmenite, and Ti-hematite, and Ce in allanite and monazite.

Zircon is likely the most problematic contaminant phase, given the abundance of petrographically observed inclusions. Sharp spikes in laser-ablation downhole Zr counts were observed for analyses of sample OM20-13 and OM20-17, such that analyses with sufficient stable signal Zr are interpreted to be free from zircon contamination (See Table S4). In some cases, no spikes were observed, whereas the analytical window was restricted to the stable portions of the signal during data processing for other analyses. Measured background garnet Zr contents are 1.03–1.30 μg/g, 0.72–0.78 μg/g, and 0.32–0.86 μg/g in samples OM20-13, OM20-17, and OM20-18, respectively. These data contrast with the 3–4 orders of magnitude higher Zr contents measured in most LASS-ICPMS analyses (see below).

Measured Ti garnet contents show compositional ranges of 117–422 μg/g and 89–243 μg/g in samples OM20-13 and OM20-17, respectively. In sample, OM20-18 a transect shows a decrease from 567 μg/g Ti in the core to 164 μg/g Ti in the rim. Therefore, variable Ti contents may correspond to real compositional zoning and not uniquely identify contamination by a Ti oxide phase. In contrast, garnet Nb contents range from below a detection limit of 8.9 ng/g to a maximum of 1.5 μg/g. The transect in sample OM20-18 shows a core to rim decrease Nb from 1.00 μg/g to below detection, mirroring the zoning observed in Ti. Whereas Ti is a minor element in garnet, we show that Nb contents are consistently low in garnet but are likely much higher in Ti oxide phases. For example, Garber et al. (2021) measured Nb contents of rutile in the range 250 to 9000 μg/g from As Sifah eclogites. We calculate $Nb_{Rt}/Nb_{Grt}$ ratios on the order of $10^4$ to $10^5$ (or higher for garnet analyses below detection limits), whereas $Ti_{Rt}/Ti_{Grt}$ ratios are the order of $10^3$. Therefore, Nb is a more sensitive indicator than Ti of contamination by Ti oxide phases.





Garnet Ce contents were below a calculated detection limit of 4.5 ng/g for most analyses not affected by zircon contamination, whereas spots with higher Zr also contain higher Ce (see below). In contrast, Garber et al. (2021) measured Ce

contents of up to tens of μg/g, which may reflect the influence of contaminant phases in their analyses.

Uranium, Th, and Pb contents in garnet are also low in all three samples. Measured background U contents range from 1 to 27 ng/g, with large uncertainties of 30–60 % (2s) due to low signal. Thorium contents range from below a detection limit of 0.3 ng/g to 30 ng/g. Finally, Pb contents, calculated from the [208]Pb signal, range from below a detection limit of 6.6 ng/g to a maximum of 0.17 μg/g. Comparison of U, Th, and Pb contents measured on the Neptune Plus compared to contents

of these elements measured using the Element-XR during LASS-ICPMS analyses show a systematic offset from the 1:1 line (e.g., Fig. S4 for U), with calculated U, Th, and Pb contents higher in the Neptune Plus data compared to simultaneously collected . A larger offset is observed for Session 1 compared to Session 2 and the offset is highly sensitive to the pit-depth correction. We suggest that there may be slight differences in the garnet ablation rate between sessions and samples that we do not account for by applying an average pit-depth correction. Therefore, we use the U, Th, and Pb contents measured by the

Element-XR ICPMS and processed in LADR for comparison between the LASS-ICPMS data and the background U, Th, Pb data described here. In both cases, the trace-element data was analysed using the Element-XR and data processed with LADR, providing consistency between the separate LASS-ICPMS and LA-ICPMS sessions.

### 4.3 Evidence for minor and trace element mixing between garnet and inclusions

Simultaneous U-Pb isotopic and trace element data collected by LASS-ICPMS spots on garnet shows trends that

identify widespread contamination by U- and Th-bearing inclusions. These inclusions dominate the U, Th, and Pb contents of the measured analyses in all samples. Linear mixing arrays between garnet, whose background minor- and trace-element compositions are described in Section 4.1, and zircon and rutile were calculated. Below we assess these trends on a sample-by-sample basis.

### 4.3.1 OM20-13

Linear mixing arrays of U vs Zr, Th vs Zr, Th vs Ce, U vs Nb, Nb vs Ti, and Nb vs. Zr between garnet and potential contaminants of the U-(Th-)Pb isotopic system are shown in Figure 4 for sample OM20-13. Trends in U vs Zr and Th vs. Zr display positive linear correlations consistent with mixing between garnet (average Zr = 1.17 μg/g, U = 17 ng/g, Th = 12 ng/g) and zircon (49.77 wt. % Zr for pure ZrSiO₄) with 1000 to 2400 μg/g of U and 200 to 1600 μg/g of Th (Fig. 4a and 4b). Some analyses show elevated Th and Ce contents and high Th and at low Zr, consistent with contamination by a Th + REE enriched

phase such as monazite or allanite (Fig. 4b and c). The plot of U vs Nb (Fig. 4d) shows that most data exhibit variable U (0–2.5 μg/g) at Nb ≤6 μg/g. Mixing lines between garnet (Nb = 0 μg/g and U = 17 ng/g) and rutile (assuming an average Nb of 500 μg/g calculated from Garber et al., 2021) are plotted for U contents of 50 to 350 μg/g in rutile, but these potential mixing lines only intersect a small portion of the data array with both elevated Nb and U. A plot of Nb vs Ti displays analyses with



200–400 µg/g at low Nb, likely representing Ti zoning in garnet, with a linear array of analyses extending to high Nb and Ti.
LA-ICP-MS analyses of garnet from OM20-13 that are free from Ti spikes display tens of ng/g or less of Nb (Table S4),
requiring inclusion contamination to describe the linear Nb vs Ti array in Figure 3e. This array shows some dispersion,
consistent with mixing of garnet with variable Ti and rutile with variable Nb. Finally, a plot of Nb vs Zr shows little correlation
between these elements (Fig. 4f).

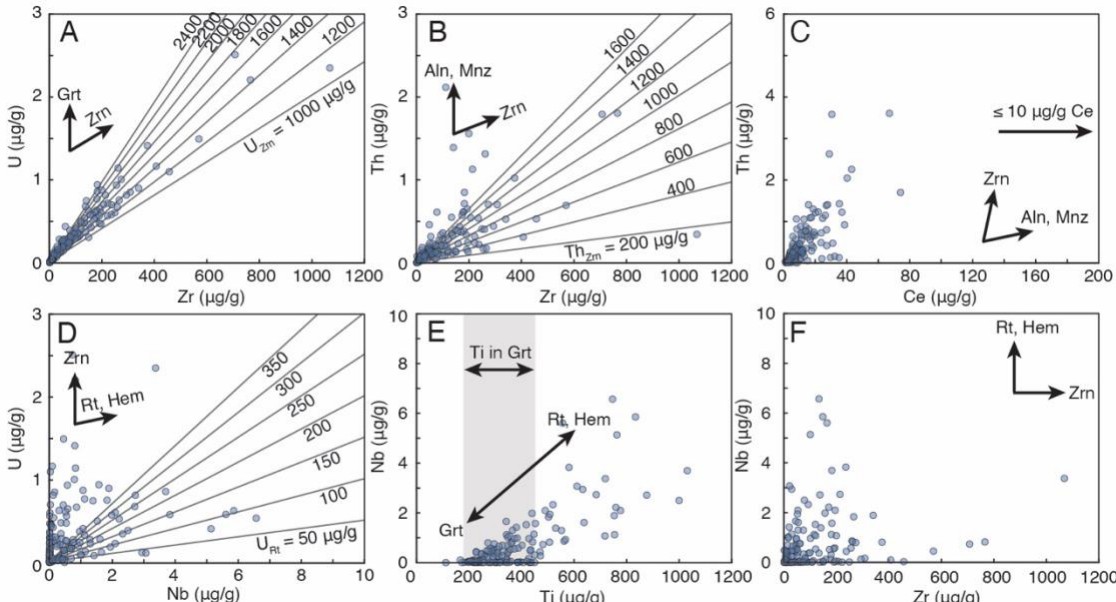


**Figure 4: Plots of U vs Zr (a), Th vs Zr (b), Th vs Ce (c), U vs Nb (d), Nb vs Ti (e), and Nb vs Zr (f) for LASS-ICPMS analyses of garnet in sample OM20-13. Mixing lines between garnet and zircon and rutile are plotted for a range of U and Th contents. Mineral abbreviations after Warr (2021).**

### 4.3.2 OM20-14

420        Plots of U vs Zr and Th vs Zr (Fig. 5a and 5b) show linear trends consistent with mixing between garnet (Zr = 1 µg/g,
U = 0 µg/g, Th = 0 µg/g) and zircon with 50–600 µg/g U (average ≈ 300 µg/g U) and 20–160 µg/g Th, respectively. The garnet
background Zr, U, and Th contents were not determined for OM20-14; however, garnet Zr contents measured in the three
samples in which 33 µm diameter LA-ICPMS analyses were conducted were consistently low (0.5–1.5 µg/g) and U and Th
contents were at low ng/g level. Therefore, assumptions regarding the lower intercept of the mixing arrays are unlikely to
significantly change their slopes. In the plot of Th vs Ce, (Fig. 5c), eight analyses plot at elevated Ce (10–100 ng/g Ce) and
moderate to high Th (3–30 ng/g), whereas most analyses fall along a linear trend projecting to high Th at low Ce. The plot of
U vs Nb (Fig. 5d) shows data extending to both high Nb and low U, falling within an array consistent with mixing between
garnet (U = 0 ng/g and Nb = 0 µg/g) and rutile (Nb = 500 µg/g) with U contents of 10 to 60 µg/g. However, most analyses plot




outside of this array. The plot of Nb vs Ti shows a cluster of analyses at low Nb and 100–500 μg/g Ti, like analyses from

OM20-13, and are interpreted to represent Ti zoning in garnet (Fig. 5e). A well-defined linear array of Nb vs Ti far exceeds background garnet Ti and Nb contents measured for samples OM20-13, OM20-17, and OM20-18. A simple least squares regression shows that this trend is well-defined, with an $R^2$ of 0.95. Assuming this trend represents mixing between garnet and rutile, the slope of the regression projects to a Nb content of ~460 μg/g for rutile, consistent with the average Nb content of rutile (~500 μg/g) measured by Garber et al. (2021) in rutile from As Sifah. The plot of Nb vs Zr show a cloud of data covering

nearly the entire range of potential Nb/Zr ratios (Fig. 5f), with most clustered at relatively low Nb and Zr.

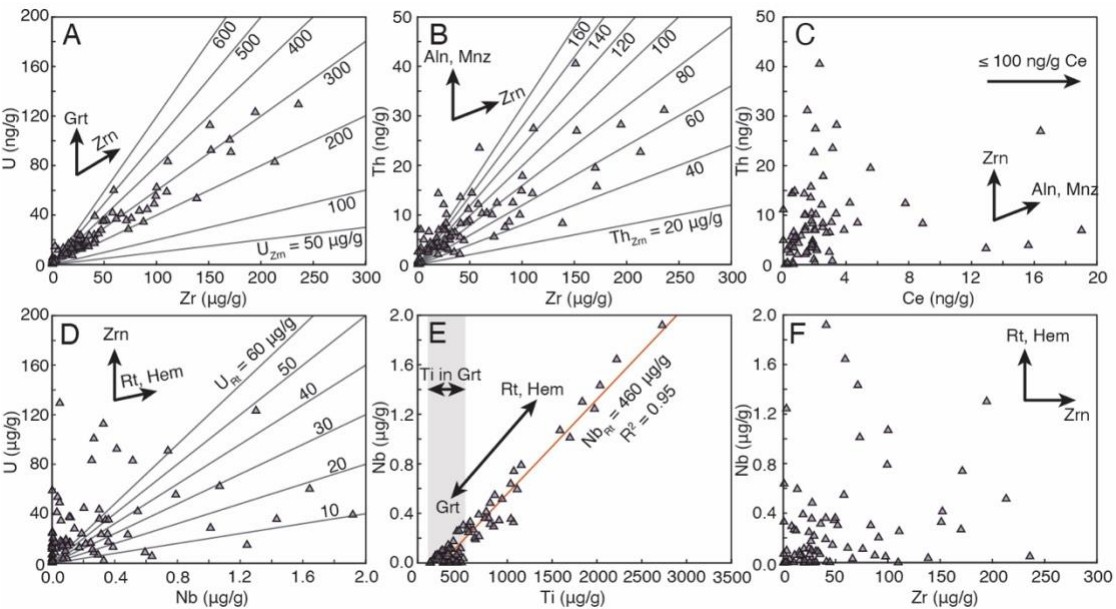

**Figure 5: Plots of U vs Zr (a), Th vs Zr (b), Th vs Ce (c), U vs Nb (d), Nb vs Ti (e), and Nb vs Zr (f) for LASS-ICPMS analyses of garnet in sample OM20-14. Mixing lines between garnet and zircon and rutile are plotted for a range of U**

**and Th contents. Mineral abbreviations after Warr (2021).**

### 4.3.3 OM20-17

Plots of U vs Zr and Th vs Zr display linear positive correlations in these elements (Fig. 6a and 6b). These trends fall on mixing lines between garnet (average Zr = 0.747 μg/g, U = 1.7 ng/g, Th = 0.6 ng/g) and zircon with 200–1000 μg/g U and 100–700 μg/g Th. Nearly all analyses fall on a mixing line with zircon containing 400–600 μg/g U. The plot of Ce vs Th (Fig.

6c) shows that only one analysis falls above 10 ng/g Ce, whereas all other data fall between 0 and 0.25 μg/g Th at 0–8 ng/g Ce. In the plot of U vs Nb, 63 % of analyses cluster at <0.2 μg/g Nb and <0.05 μg/g U, with most of the remaining analyses plotting between 0 and 0.15 μg/g U and up to ~1.8 μg/g Nb (Fig. 6d). Similar to sample OM20-14, some of the low U–high Nb analyses fall between mixing lines for garnet (U = 1.7 ng/g and Nb = 0 μg/g) and rutile (Nb = 500 μg/g) with U contents



of 20 to 140 µg/g. Low Nb analyses plot between 100 and 300 µg/g Ti, whereas analyses up to ~1450 µg/g Ti plot along a
linear array in Nb vs Ti space (Fig. 6e). Assuming the linear regression of these data ($R^2 = 0.96$) represents rutile contamination,
the estimated average Nb content of the rutile is 860 µg/g. Also, like in OM20-14, the plot of Nb vs Zr shows a cloud of data
with elevated Nb and Zr, but without a clear trend (Fig. 6f).

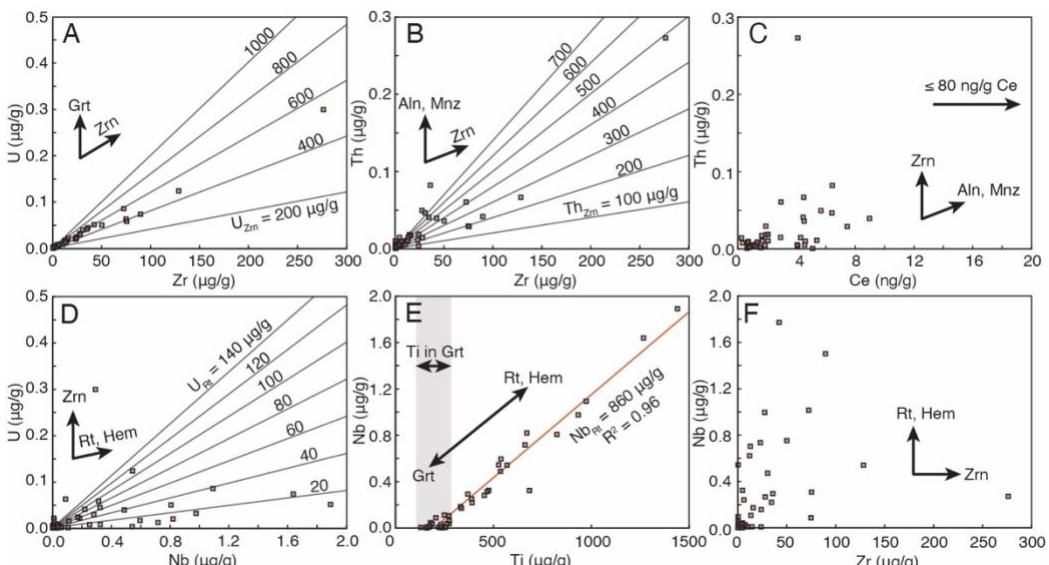

**Figure 6: Plots of U vs Zr (a), Th vs Zr (b), Th vs Ce (c), U vs Nb (d), Nb vs Ti (e), and Nb vs Zr (f) for LASS-ICPMS
analyses of garnet in sample OM20-17. Mixing lines between garnet and zircon and rutile are plotted for a range of U
and Th contents. Mineral abbreviations after Warr (2021).**

### 4.3.4 OM20-18

Analyses in Sample OM20-18 also plot along linear arrays in U vs Zr and Th vs Zr space (Fig. 7a and 7b). These data
are largely bounded by mixing lines between garnet (average Zr = 0.558 µg/g, U = 12.7 ng/g, Th = 3 ng/g) and zircon with
1200–2400 µg/g U and 100–700 µg/g Th. In the plot of Ce vs Th, 90 % of analyses plot at Ce <0.1 µg/g and Th contents of 0
to almost 0.15 µg/g (Fig. 7c). The remaining analyses plot at both elevated Th and Ce contents, indicating contamination by a
REE-rich phase, such as monazite or allanite. Similarly, 88 % of analyses in the U vs Nb diagram plot below 1 µg/g Nb but
exhibit variable U (Fig. 7d). The highest Nb contents (up to 11 µg/g) plot at low U, consistent with background garnet U
contents. Only four analyses with high U (out of 133 analyses) plot on mixing lines between garnet (U = 12.7 ng/g and Nb =
0 µg/g) and rutile (Nb = 500 µg/g). In Nb vs Ti space (Fig. 7e), low Nb–high Ti analyses fall between 200–500 µg/g Ti,
consistent with measured Ti zoning measured in a core–rim transect from OM20-18 using 33 µm spots (see Section 4.1). A
positively correlated array of data extends from low Nb and background Ti contents to a maximum of ~11.5 µg/g Nb and




~1330 μg/g Ti. Finally, Nb and Zr are decoupled, with the highest Zr contents occurring at low Nb and the highest Nb contents occurring at low Zr (Fig. 7f).

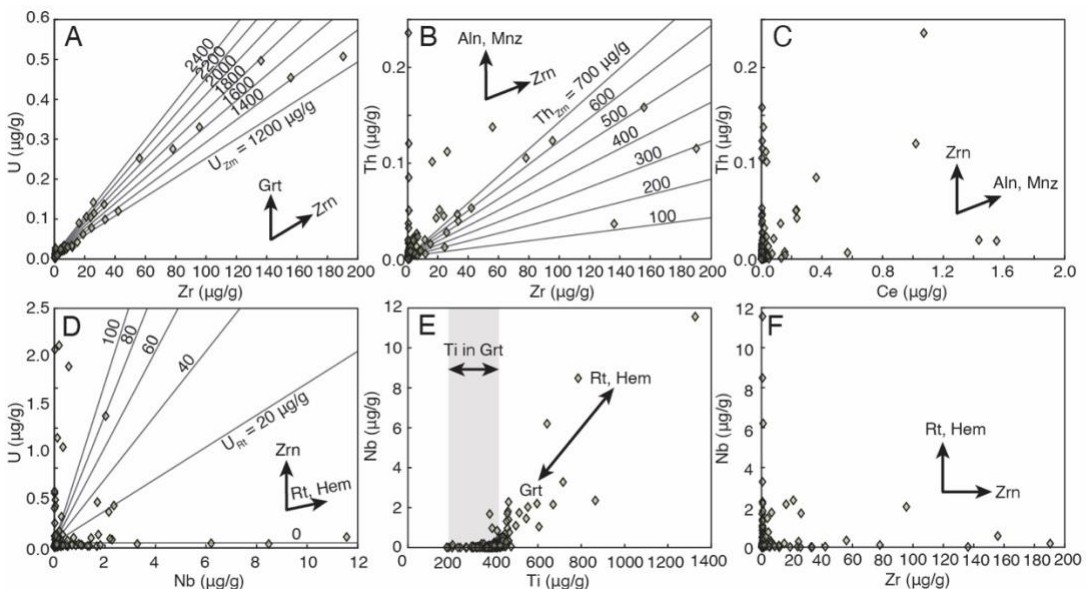

**Figure 7: Plots of U vs Zr (a), Th vs Zr (b), Th vs Ce (c), U vs Nb (d), Nb vs Ti (e), and Nb vs Zr (f) for LASS-ICPMS analyses of garnet in sample OM20-18. Mixing lines between garnet and zircon and rutile are plotted for a range of U and Th contents. Mineral abbreviations after Warr (2021).**

### 4.3.5 CWO21

Analyses of garnet in CWO21 display a positive correlation in U vs Zr (Fig. 8a). These data are bounded by mixing lines between garnet (Zr = 1 μg/g, U = 0 μg/g) and zircon with 200 to 1600 μg/g U. The plot of Th vs Zr shows less linearity, but the data are generally positively correlated and are bounded by mixing lines between garnet (Zr = 1 μg/g, Th = 0 μg/g) and zircon containing 50 to 600 μg/g Th (Fig. 8b). Measured Ce and Th contents are <1 and <1.5 μg/g (Fig. 8c), respectively, with analyses trending to either high Th at low Ce or low Th and high Ce. In the plot of U vs Nb, 56 % of analyses cluster at ≤1 μg/g Nb and ≤0.05 μg/g U (Fig. 8d). The remaining data extend to ~0.4 μg/g U at low Nb or show both elevated U and Nb, where the latter are bounded by mixing lines between garnet (U = 0 ng/g and Nb = 0 μg/g) and rutile (Nb = 500 μg/g) with 5 to 30 μg/g U. Analyses which display low Nb contents exhibit Ti contents between 150 and 650 μg/g, likely representing Ti zoning in garnet (Fig. 8e). The remaining data extend in a linear array up to ~7 μg/g Nb and 6500 μg/g Ti. If this trend represents mixing between garnet and rutile, the linear regression of this trend ($R^2$ = 0.99) projects to a potential Nb of 665 μg/g for rutile. Finally, the plot of Nb vs Zr shows decoupling between these elements, such that analyses exhibit a range of Zr contents between 0 and 190 μg/g regardless of Nb content (Fig. 8f).





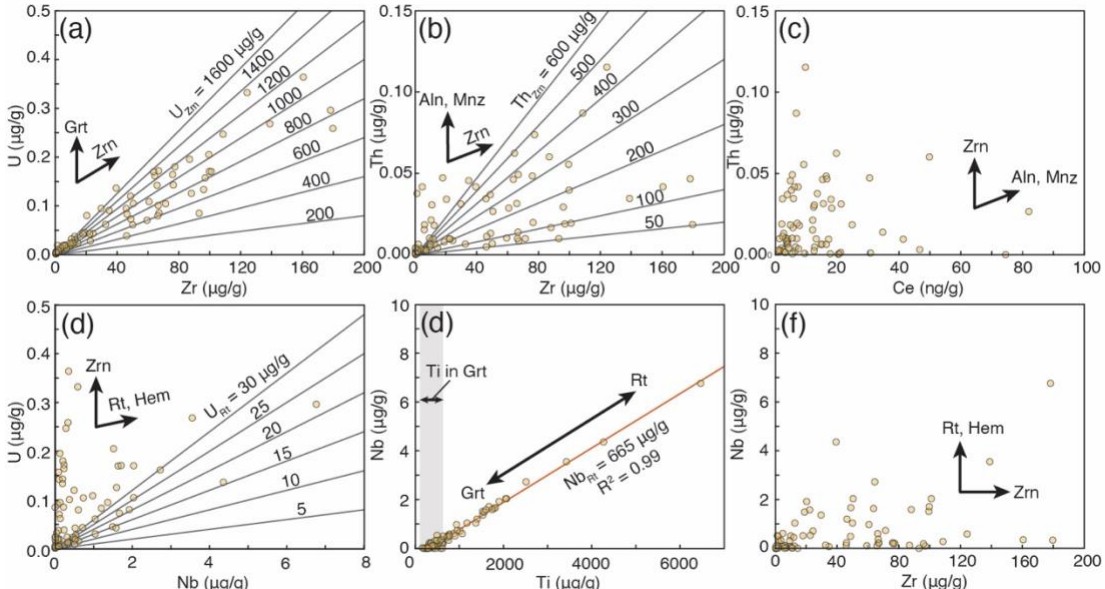

**Figure 8: Plots of U vs Zr (a), Th vs Zr (b), Th vs Ce (c), U vs Nb (d), Nb vs Ti (e), and Nb vs Zr (f) for LASS-ICP-MS**
**analyses of garnet in sample CWO21. Mixing lines between garnet and zircon and rutile are plotted for a range of U**
**and Th contents. Mineral abbreviations after Warr (2021).**

**4.4 U-Pb geochronology**

The LASS-ICPMS U-Pb isotopic data are plotted in Tera-Wasserburg diagrams in which data are aligned along regression lines, which show linear mixing between common Pb (y-intercept) and ingrown radiogenic Pb (Fig. 9). The data in
Figure 9a-e were screened for sharp spikes or disruptions in the downhole U and Pb signals but trace-element downhole signals and contents were not considered at this stage. A $2\sigma$ filter was applied on the data during calculation of the lower intercept ages; however, most analyses fall within the $2\sigma$ uncertainty envelope of the regression line and calculated mean squares of the weighted deviates (MSWD) of 0.97 to 1.93 reflect that the calculated ages are statistically well-defined by the remaining analyses. Calculated concordia intercept dates range from $88.8 \pm 1.9$ to $94.4 \pm 7.2$ Ma. Although there is some variability, all
analyses with elevated U contents and high $^{238}U/^{206}Pb$ co-occur with Zr contents that are elevated above background garnet Zr contents (see previous section). Garnet U-Pb error envelopes on individual data points are linearly color-coded in grayscale as a function of Zr content, between background (white) and the highest Zr content for each sample (black). While there is some spread in the color-coded data, analyses with the lowest Zr contents consistently plot near the y-intercept with common Pb (Fig. 9).



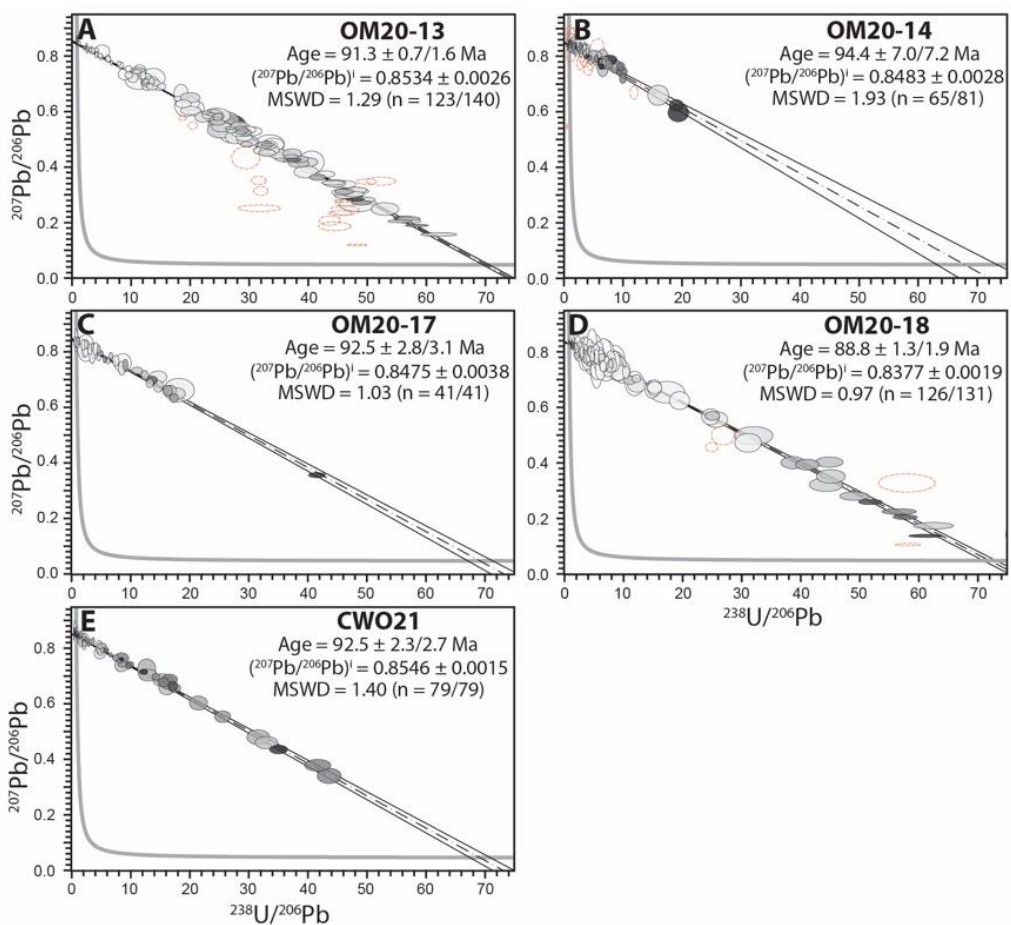


**Figure 9: Tera-Wasserburg diagrams of garnet U-Pb LASS-ICPMS analyses for OM20-13 (a), OM20-14 (b), OM20-17 (c), OM20-18 (d), and CWO21 (e). Data points are color-coded in grayscale for Zr content between white (lowest Zr content) and black (highest Zr content). Red dashed analyses show data that are not included in the regression. The black dashed line is the isochron, whereas the solid black lines are the 2s uncertainty envelope and thick gray line is**

**concordia.**

## 5 Discussion

If the 94–89 Ma garnet U-Pb dates calculated here accurately constrained the timing of eclogite-facies metamorphism then our data would require, 1. a reason for decoupling between U-Pb and Sm-Nd isotopic systems under relatively low-*T* metamorphic conditions (~550 °C and lower), and 2. a tectonic explanation for simultaneous lawsonite eclogite-facies

metamorphism of the As Sifah eclogite and granulite-facies metamorphism of the sole of the Semail Ophiolite. Garnet U-Pb dates of 94–89 Ma are significantly older than 81–77 Ma ages calculated by Sm-Nd garnet, U-Pb zircon, and U-Pb rutile geochronology for the As Sifah eclogite (Warren et al., 2003; Warren et al., 2005; Garber et al., 2021). In particular, Garber et



al. (2021) determined a Sm-Nd garnet age of 77.5 ±2.2 Ma for sample CWO21, whereas the U-Pb 'garnet' date calculated here for CWO21 is 92.5 ±2.7 Ma. Additionally, the 94–89 Ma garnet U-Pb dates calculated here overlap (within uncertainty)

the formation age of the oceanic crust of the Semail ophiolite (Rioux et al., 2012; 2013; 2021) and granulite-facies subduction metamorphism of the metamorphic sole at ca. 95 Ma (Hacker et al., 1996; Warren et al., 2005, Rioux et al., 2016; 2023), which structurally overlie the high-*P* rocks of the Saih Hatat window. Instead of a geological explanation, we suggest that these apparent discrepancies can be resolved analytically: Our garnet U-Pb data do not constrain the timing of lawsonite eclogite-facies metamorphism, but instead the U-Pb regression lines represent mixing between dominantly unradiogenic common Pb

in garnet and radiogenic Pb and U in zircon micro-inclusions.

### 5.1 Evidence for zircon contamination

Plots of U vs. Zr show positive linear correlations in all samples (Figs. 4–8). These trends are consistent with linear mixing between garnet and zircon during ablation. Linear Zr-U trends observed here (Figs. 4–8) project to high Zr at low U. These trends are inconsistent with any coupled substitution of U and Zr into the garnet structure and instead plot towards U

and Th contents that are reasonable for zircon. Additionally, the 33 μm trace element analyses, which were conducted by selecting every spot location optically and by BSE for areas free of zircon inclusions before analysis, have uniformly low Zr and U compared to the larger-spot LASS-ICPMS analyses (see results). Finally, a close inspection of the samples reveals that zircon micro-inclusions (≤2 μm) are ubiquitous in garnet grains in all five samples analysed in this study (Fig. 2). These observations are consistent with trace-element garnet maps (Fig. 3), which show that elevated U and Zr contents co-occur in

discrete pixels, whereas the surrounding garnet contains little U and Zr. The density of zircon micro-inclusions in garnet in all samples, except OM20-18, suggest that co-ablation of zircon inclusions is unavoidable given the large ablation spot size (193 μm diameter) necessary to achieve sufficient U and Pb counts for age dating.

The LASS-ICPMS screening method is highly sensitive to the detection of co-ablated zircon considering the low garnet background Zr content (0.5–1.5 μg/g) relative to the Zr content of zircon (49.77 wt %). For example, ablation of a single

2 μm wide spherical zircon inclusion is sufficient to raise the Zr content of the analysis from 1 μg/g in pure garnet to 9 μg/g of Zr, a scenario in which 99.998 vol. % of the ablated material is garnet and only 0.002 vol. % is zircon. Using the background corrected Zr contents measured by LASS-ICPMS and the densities of zircon (4.71 g/cm$^3$) and garnet (4.31 g/cm$^3$), we can similarly calculate the volume of zircon ablated in each analysis. For OM20-13, OM20-17, and OM20-18, the average background garnet Zr contents were measured using a 33 μm laser spot size (Section 4.2), whereas a background content of

0.5 μg/g is assumed for OM20-14 and CWO21. Zircon is estimated to contribute from 0 to a maximum of 2 vol. ‰ to each LASS-ICPMS analysis (see Table S5). Therefore, if U contents depend on zircon contamination in our samples, then very little contamination is required to reach the elevated Zr (and U) contents observed in our analyses relative to those of pure garnet in our samples.





If our isochrons are defined by inclusion contamination, then there must be a relationship between the volume of
zircon ablated and the U and Pb isotopic composition measured. Data plotted in the Tera-Wasserburg diagrams in Figure 9 are
color-coded for Zr content, showing that in all samples low Zr contents plot near the y-intercept of the isochrons. As a result,
these analyses must exhibit a Pb isotopic composition near to that of common unradiogenic Pb. However, the highest Zr
content datapoints plot over a range of $^{207}$Pb/$^{206}$Pb and $^{238}$U/$^{206}$Pb ratios, which may be the result of varying U contents in the
ablated zircon inclusions. To test this hypothesis, we plotted the calculated volume of ablated zircon vs. the percentage of
radiogenic Pb for each sample and color-coded the data by their measured U contents (Fig. 10). The percentage of radiogenic
Pb is calculated for each datapoint by linear mixing between concordia and common Pb. While there is significant spread in
the data, each sample displays a positive correlation between U content, percentage of radiogenic Pb, and volume of ablation
zircon. While all minerals will develop increased radiogenic Pb with higher U contents, the correlation with the volume of
ablated zircon is consistent with co-ablated zircon inclusions in our dataset.  Combined with all other lines of evidence (e.g.,
frequent zircon micro-inclusions, discrete U and Pb hot zones in the LA-ICPMS map, and shallow Zr:U mixing trends), suggest
that the U-Pb dates calculated here (Fig. 9) reflect the crystallisation age of zircon micro-inclusions rather than eclogitic garnet.





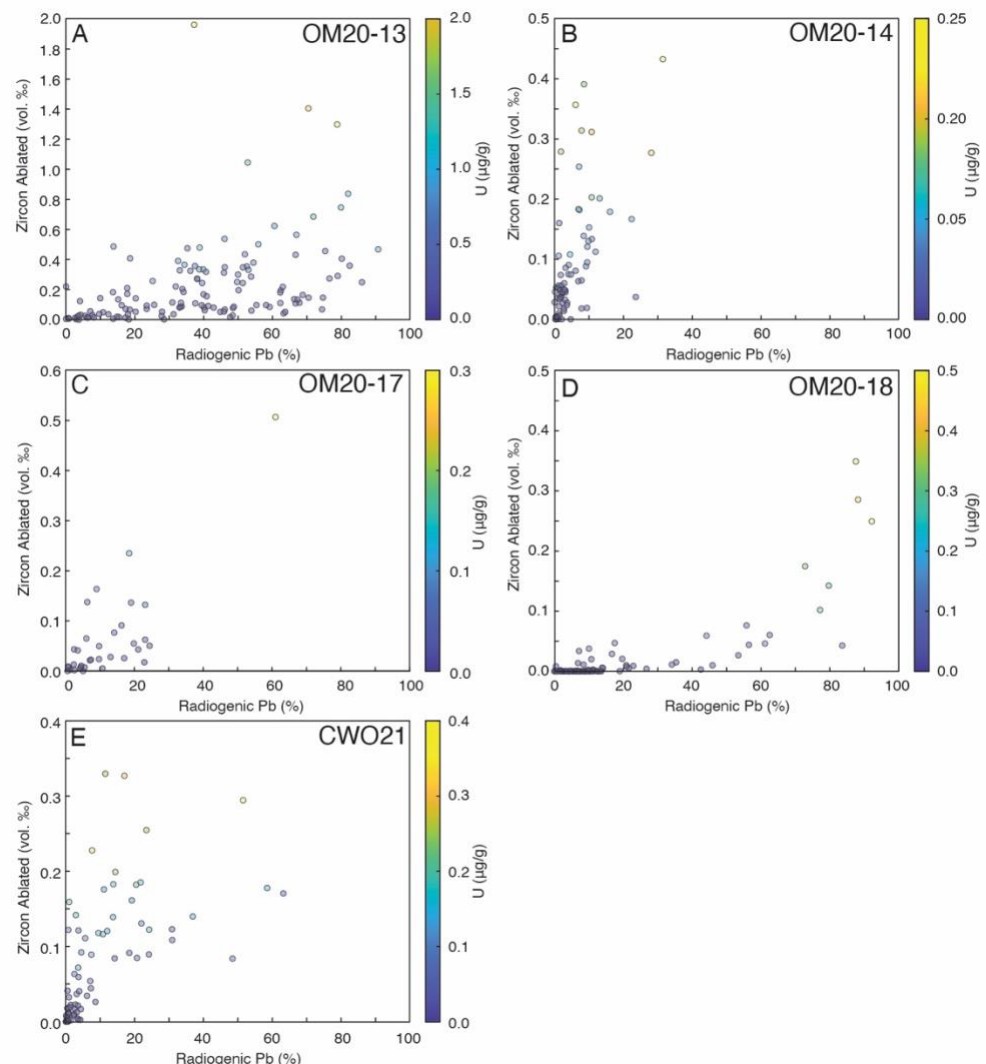

**Figure 10: Plots of volume of zircon ablated vs the percentage of radiogenic Pb for OM20-13 (a), OM20-17 (c), OM20-18 (d), and CWO21 (e). Data are color-coded by U content.**


Contributions from other U-bearing phases to the measured U and Pb isotopes are limited. For example, relatively few analyses plot at both elevated Ce and Th, and no linear correlations consistent with mixing are observed (Figs. 4–8). We did not petrographically observe monazite, allanite, or other REE-enriched phases that may contain elevated U (and Th); therefore, significant contamination of the U-Pb system by REE mineral inclusions is unlikely. In contrast, rutile and ilmenite

(±Ti-hematite) are observed in all samples and may contain both U and Pb. Strong linear correlations in Nb vs. Ti are observed and the many analyses far exceed background contents of these elements measured on 'clean' garnet. These correlations project to potential Nb contents of rutile from 460 to 860 μg/g in rutile, falling within the range of Nb contents measured by Garber



et al. (2021) for rutile from As Sifah eclogite. Rutile U contents measured by Garber et al. (2021) do not exceed 30 μg/g and most analyses were <10 μg/g U. Mixing lines between garnet and rutile in U vs. Nb space show that, the U contents of many

analyses are too high and Nb contents too low for rutile to have contributed the majority of U, even though some data are consistent with mixing between garnet and rutile with ≤30 μg/g U. Rutile contamination also cannot account for the linear correlation in U and Zr. The Zr contents of rutile are 20–40 μg/g (Garber et al., 2021), whereas higher contents are frequently measured here. Mixing lines between garnet and rutile in Zr vs U space would also plot at much steeper slopes than garnet-zircon mixing lines, inconsistent with the shallow slopes of the Zr vs U trends. Additionally, no correlation is observed between

Nb and Zr, suggesting that rutile contamination makes little contribution to the Zr content. Since U and Zr are coupled, but Nb and Zr are not, it is not possible for contamination by rutile or other Ti-oxide phase to account for the elevated U contents of our analyses. Therefore, data plotting between garnet and rutile mixing lines in U vs Nb are likely due to co-ablation of zircon and rutile instead of a significant U contribution from rutile alone, resulting in a significant number of analyses with elevated Zr and Nb.

**5.2 Best practices in screening for contamination**

Since the first application of in situ garnet U-Pb geochronology by LA-ICPMS by Seman et al. (2017), most studies have focused on dating grossular–andradite series garnets, which typically contain μg/g levels of U (e.g., DeWolf et al., 1996; Seman et al., 2017; Wafforn et al., 2018; Burisch et al., 2019; 2023) and few high-U inclusions, if any. Recent studies by Millonig et al. (2020), Schannor et al. (2021), Mark et al. (2023), Bartoli et al. (2024), and Shu et al. (2024) have expanded U-

Pb geochronology by LA-ICPMS to common metamorphic garnet with very low U contents (<1 μg/g). Early ID-TIMS studies of the U-Pb system in garnet by Mezger et al. (1989) and DeWolf et al. (1996) suggested that, whereas U is systematically incorporated at μg/g levels into garnet during growth in Ca-rich skarn-type garnets, the U contents of inclusions in metamorphic almandine-pyrope series garnet can be vastly higher than their host. Inclusion-free analyses of eclogitic garnet in our study display U contents of 1 to 20 ng/g. These contents are lower than those in almandine-pyrope series metamorphic

garnet dated by Millonig et a. (2020). At such low U contents, we show that very little inclusion contamination is required to completely overwhelm the garnet U (and Pb) signal (typically <1 vol. ‰).







**Figure 11: Tera-Wasserburg diagrams of garnet U-Pb LASS-ICPMS analyses for OM20-13 (a, b), OM20-14 (c, d), OM20-17 (e, f), OM20-18 (g, h), and CWO21 (i, j). Only analyses with Zr contents at or below the average Zr background content are plotted in (f). Data points are color-coded in grayscale for Zr content between white (lowest**



**Zr content) and black (highest Zr content). Red dashed analyses show data that are not included in the regression. The black dashed line is the isochron, whereas the solid black lines are the 2s error envelope and thick gray line is the concordia.**

Our data were re-examined in a double-blind exercise to test whether the inclusions may be screened entirely by looking at the U-Th-Pb data. Analyses were reprocessed by a co-author (L.M.) and checked for inclusions using U, Th, and Pb contents, Th/U ratio, and downhole MC-ICP-MS signals following Millonig et al. (2020). Importantly, these data were reprocessed without access to the split-stream trace element data. During this process, some data which contained somewhat variable downhole signals (e.g., with broad-low frequency peaks) that were likely to be garnet but may possibly show inclusion

contamination were retained. Similarly, the split-stream trace element data were reprocessed by the lead author (J.W.) without access to the U-Pb data. In both cases only the most stable portions of the signal were retained and regions of the downhole signal that were chaotic or showed obvious spikes were removed from consideration. Analyses were rejected outright as inclusion contaminated if the entirety of the signals were too unstable. For example, most analyses of samples OM20-14 and CWO21 that most spots did not have sufficiently stable signals with low Zr, Nb, Pb, Th, and U to select inclusion free segments,

thus the data for these elements in Table S6 are equivalent to those in Table S3. It is critical to note that this approach is an inherently qualitative and subjective screening process, but provides a first-order check of the reproducibility of our reduction approach. Following reprocessing and careful checking of the downhole signals, analyses which still exceeded the Zr and Ce contents of the garnet background were rejected as inclusion biased.

   The results of this double-blind study are given in Figure 11 and in Table S6, where the first column of Tera-

Wasserburg diagrams were only screened using the MC-ICPMS data. While significantly fewer data are plotted compared to Figure 9, the regression and calculated date for sample OM20-13 are largely unchanged (except for larger uncertainties). However, insufficient analyses remain after screening using the MC-ICPMS data to calculate meaningful concordia intercept ages for OM20-14, OM20-17, and CWO21. In all cases, the analyses rejected by examination of the MC-ICPMS data were also rejected during independent examination of the trace element data. However, additional analyses that appeared irregular

but were qualitatively passable during the assessment of the MC-ICPMS data were later rejected by considering cut-offs based on the background contents of the trace elements. As a result, < 10 analyses are retained for all samples except for OM20-18 following both screening procedures. For these samples, the remaining analyses are too few and are restricted to high $207Pb/206Pb$ and low $238U/206Pb$, preventing the calculation of meaningful concordia intercept ages. Only sample OM20-18, which optically and in BSE was found to qualitatively have the fewest inclusions, has enough clean garnet U-Pb analyses

and significant enough spread in U-Pb ratios to calculate a statistically meaningful concordia-intercept date. For this sample, only 8 additional analyses are rejected following both screening processes. The calculated date of 72 ±7 Ma is significantly younger than the 89 ±2 Ma date calculated for the same sample when high Zr analyses are also considered. The date is also significantly younger than the dates of all other samples prior to the double-blind screening exercise. However, 72 ±7 Ma is consistent within uncertainty of the 81–77 Ma Sm-Nd garnet ages of Garber et al. (2021) for As Sifah eclogite samples.




In our double-blind test, we discovered that most of the inclusion-affected analyses were identified by examining the U, Th, and Pb MC-ICPMS data alone; however, tens of analyses still showed zircon contamination after screening. In Figure 12, we show examples of the time-resolved $^{238}$U and $^{208}$Pb signals (in cps) for andradite garnet reference Mali Black (Fig. 12a), a low-Zr and low-radiogenic Pb garnet analysis from sample OM20-18 (Fig. 12b), and a high-Zr high-radiogenic Pb inclusion contaminated garnet analysis from sample OM20-13 (Fig. 12c). The reference material analysis shows a flat stable signal in

$^{238}$U and $^{208}$Pb, whereas the analysis of low-Zr garnet from OM20-18 shows a noisy, but relatively stable, signal for $^{238}$U at very low counts. Despite the elevated Zr content of analyses from OM20-13, there are no clear spikes in $^{238}$U and $^{208}$Pb that would indicate the ablation of distinct zircon inclusions (Fig. 12c). Instead, many high-Zr analyses show somewhat irregular downhole $^{238}$U and $^{208}$Pb signals, but without the sharp spikes that are commonly indicative of the ablation of individual inclusions. In some analyses areas of high $^{238}$U and $^{208}$Pb are observed; however, elevated U and Pb counts occur as low and

broadened humps. Whereas relatively large individual zircon inclusions may show up as sharp spikes in the time-resolved U, Th, and Pb signals, here the zircon micro-inclusions contribute a very small volume to the overall signal (<1 vol. ‰). In our analyses, inclusion dominated U, Th, and Pb signals are relatively smoothed due to the combination of the high frequency of zircon inclusions, small inclusion size, ultra-low U content of garnet, and large ablation spot size (193 µm). However, a rejection of all but the most stable signals (e.g., like those in Fig. 12c & d) produces identical results to the detailed screening

for inclusions in our double-blind test. We therefore suggest that a conservative approach would be the total rejection of analyses that exhibit sloped and/or moderately unstable downhole signals, if split-stream data are not available.



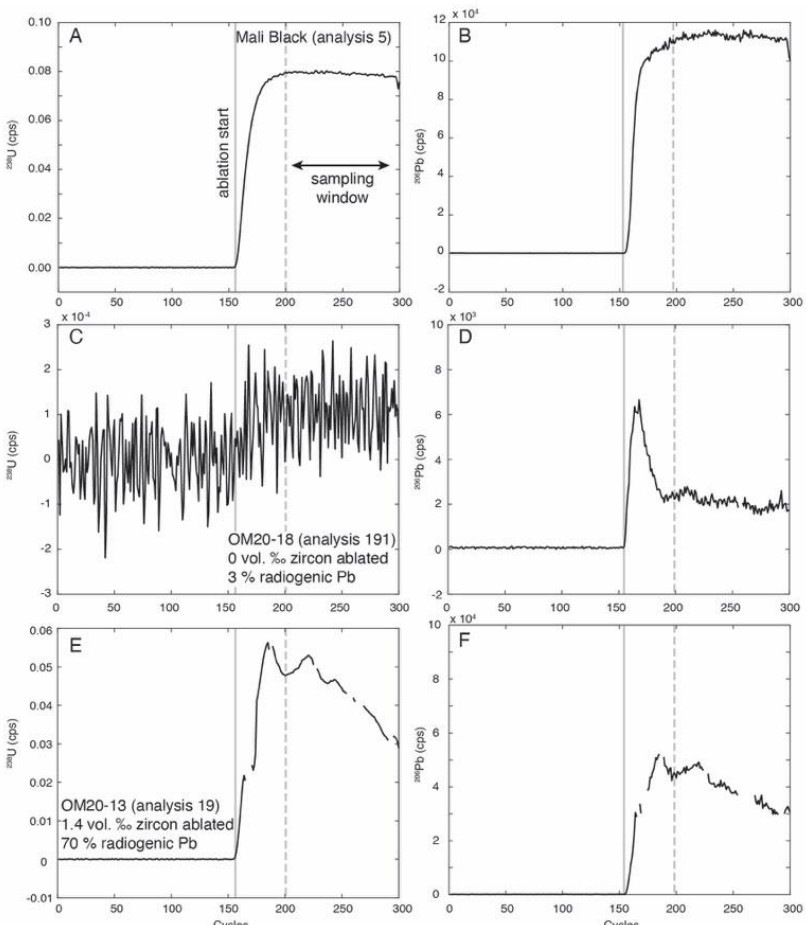

**Figure 12: Time resolved $^{238}$U and $^{208}$Pb down hole signals (in counts per second, cps) for the Mali Black reference material (a, b), OM20-18 (c, d), and OM20-13 (e, f). Some electronic detector noise was removed (d, e, f). Note, that whereas the U signal of (c) is noisy, the noise in $^{238}$U and $^{206}$Pb is somewhat correlated, such that there is significantly lower noise if $^{238}$U/$^{206}$Pb is plotted. Additionally, the mean $^{238}$U/$^{206}$Pb and associated standard error over the time interval is propagated into the intercept date calculation, resulting in lower uncertainties than one might at first expect from such a noisy signal.**

      The use of U and Th contents and Th/U ratios measured on inclusion-free garnet as a cut-off remains a useful tool for identifying inclusion contamination (e.g., Millonig et al., 2020); however some problems may arise. First, the U content of inclusion-free garnet is likely variable among different samples, meaning that a single universal U cut-off content should not be applied to all garnets in all samples and instead should be determined on a sample-by-sample basis. The locations for these analyses should be carefully selected optically and by scanning-electron microscope to minimize possible contamination. Second, the U content should be determined using the same analytical routine, instrument, and data reduction scheme to prevent possible biases. However, we demonstrate that simultaneous collection of U-Pb isotopic and trace-element data by LASS-



ICPMS allows for more in-depth screening by expanding the compositional space considered. The Zr vs U trends shown here can only be explained by the frequent ablation of micro-zircon inclusions in all five of our samples. Additionally, Zr is a more sensitive element to use to define a cut-off criterion compared to U in our samples, given Zr is a major element in zircon (wt % level) whereas U typically occurs as a minor to trace element (hundreds to thousands of µg/g in most cases). Other elements

may be more useful in high-*T* or Ti-rich garnets where Zr contents in garnet may be tens or hundreds of µg/g. In such cases other elements, such as Hf, may be more useful than Zr. Regardless, LASS-ICPMS is a powerful tool for *in situ* U-Pb garnet geochronology; however, similar results may be obtained by taking a conservative approach to LA-ICPMS signal processing combined with examination of U, Th, Th/U, and garnet microtextures.

## 5.3 Significance of the U-Pb dates

Our data demonstrate that the garnet U-Pb data measured here are dominated by contamination by micro-zircon inclusions. Therefore, the inclusions dated here must be coeval with or predate garnet growth. Inclusion contamination accounts for the apparent discrepancy between the 94–89 Ma U-Pb dates calculated here, and the 81–77 Ma Sm-Nd garnet–whole rock ages calculated by Garber et al. (2021). Unfortunately, for most samples only a few analyses remain after removing analyses with Zr contents elevated above the garnet background and these data are insufficient to resolve meaningful garnet

U-Pb concordia intercept ages (Fig. 11). Sample OM20-18 exhibits the fewest inclusions in garnet, both in our analyses and petrographically, and some radiogenic Pb is present in uncontaminated garnet analyses. Figure 9f shows the U-Pb isochron after removing analyses with elevated Zr contents, a procedure, which results in a concordia intercept date of 72 ±7 Ma. This date is somewhat younger than the 77–81 Ma Sm-Nd garnet–whole rock ages of Garber et al. (2021) but overlaps within uncertainty. However, large uncertainties are the result of limited variation in $^{207}$Pb/$^{206}$Pb and $^{238}$U/$^{206}$Pb at high proportions of

common Pb. Unfortunately, this date is not precise enough to resolve outstanding questions about the timing of HP-LT metamorphism at As Sifah.

The inclusion-affected analyses plot neatly along well-defined isochrons in Tera-Wasserburg space (Fig. 9), except for very few outliers. Some outliers plot along regression lines consistent with metamorphism at 77-81 Ma; however, the majority of analyses plot along regression lines consistent with 94-89 Ma intercept ages. These data suggest that the zircon

inclusions in garnet formed over a narrow window in time. Previous studies have dated zircon from the eclogites of As Sifah to 82–78 Ma (Warren et al., 2003; Gray et al., 2004; Garber et al., 2021), inconsistent with the 94–89 Ma U-Pb dates calculated here for the vast majority of zircon inclusions in garnet. Zircon grains separated by these studies are idioblastic, elongated along the c-axis, and are 100–200 µm in length, whereas the minute zircon crystals included in garnet are typically ≤2 µm in diameter, xenoblastic, and equant (see also Warren et al., 2003; 2005). Gray et al. (2004a) showed that some zircon grains

exhibit herringbone compositional zoning, possibly as the result of rapid crystallization of zircon during exhumation and suggested that the smaller zircon inclusions in garnet represent an older zircon population. If our inclusion-contaminated U-Pb data are accurate, the zircon micro-inclusion population in garnet formed at 94–89 Ma. These dates overlap with the timing



of the drowning of the Arabian passive margin (Robertson, 1987). Sediment accumulation and burial of the continental margin may have resulted in low-grade metamorphism or hydrothermal alteration in the As Sifah rocks at this time.

700          The zircon grains dated here are extremely small and the age of the ≤2 μm size fraction, and that size fraction is never measured following zircon separation. Additionally, in situ dating of individual zircon grains of this size is beyond the spatial resolution of both LA-ICPMS and secondary-ion mass spectrometry. However, micro-zircons are common in low-grade metamorphic rocks and have been interpreted to form from the breakdown of precursor minerals, such as metamict detrital zircon (Dempster et al., 2008; Hay and Dempster, 2008) and baddeleyite, pyroxene, and ilmenite (Beckman and Möller, 2018).

At low-grade conditions, reaction kinetics are slow and Zr mobility is limited, which likely results in multiple nucleation sites for zircon and small crystal size. Micro-zircon grains that are trapped in porphyroblasts are preserved, whereas others may be recrystallized during prograde metamorphism via Ostwald ripening or other recrystallization mechanisms. Such a process would produce at least two populations of zircon with distinct differences in grain size and age. Garnet does not necessarily need to grow and entrap the micro-zircon at the early stages of prograde metamorphism, instead zircon (re-)crystallization may

be overstepped and delayed until relatively high-grade conditions near the peak of metamorphism. As a result, there may be a large gap between the timing of micro-zircon crystallization and their entrapment in garnet. Therefore, our favoured interpretation of the 94–89 Ma dates obtained from the inclusion-contaminated analyses is that they reflect the timing of low-grade metamorphism or hydrothermal alteration of the Arabian margin prior to HP-LT metamorphism.

         One alternative interpretation is that the 94–89 Ma dates arise from an analytical mismatch between the NIST-

SRM614 glass and garnet U-Pb reference materials, relative to the zircon that contributed most of the measured U and Pb in the LASS-ICPMS analyses. Because of the minute volume (<1 vol. ‰) of ablated zircon in each analysis, we consider it unlikely that this would arise from laser-induced fractionation, but could result from differential plasma-induced fractionation of the zircon+garnet-laced aerosol in the ICP. One way to evaluate these geologic vs. analytical interpretations would be to separate and digest garnet with abundant zircon micro-inclusions, then analyse them for U-Th-Pb isotopes by ID-TIMS, but

this is beyond the scope of this study. However, Hollinetz et al. (2021) dated micro-zircon inclusions in chloritoid, obtaining an age consistent with K-Ar and $^{40}$Ar/$^{39}$Ar studies from the same structural unit, suggesting that bulk LA-ICPMS analysis of a low-U host and U-rich inclusions can successfully date the inclusions. Regardless, the LASS-ICPMS analyses cannot be interpreted as the timing of garnet growth in the As Sifah rocks.

## 6 Conclusions

725          In situ U-Pb garnet geochronology is a powerful tool to date a key metamorphic mineral at a high spatial resolution relative to ID-TIMS Sm-Nd and Lu-Hf garnet geochronology. Whereas grossular–andradite series and Ti-rich garnets can contain tens of μg/g of U, few studies have attempted to date common metamorphic garnets, which typically contain <1 μg/g of U. Here we show that eclogitic garnet may exhibit very low U contents (1 to 20 ng/g) and that co-ablated microscopic zircon inclusions may totally overwhelm the U signal. In many rocks, such as metasediments, igneous rocks formed by crustal partial

melting, and polymetamorphic rocks, there may not be a single inclusion population. In these cases, analyses plotted in Tera-





Wasserburg space may be scattered, such that any single regression line is not possible. However, we demonstrate that U-Pb analyses of garnet may also produce discrete well-defined regressions that are inclusion dominated and difficult to distinguish from inclusion-free garnet analyses based on U and Pb isotopic data alone. Contamination of the U-Pb system by micro-zircon micro-inclusions is nearly universal across our dataset, and it is these inclusions, which define the U-Pb regression lines, unless

very rigorous data filtering is applied. The inclusions are sufficiently small, frequent, and dispersed across their garnet host that they do not always produce clear spikes in the time-resolved U, Th, and Pb signals during analysis. However, many time-resolved signals still show slight to moderate irregularities with broad low-intensity peaks that have to be taken as indicators of co-ablation of inclusions. We demonstrate that such ambiguous analyses need to be identified by intense screening. They are clearly identified as co-ablation of zircon inclusions by examining the full LASS-ICPMS dataset. We suggest caution when

conducting in situ U-Pb garnet geochronology of inclusion-rich low U garnet if only U, Th, and Pb isotopes are measured, particularly if relatively few radiogenic analyses define the isochron. Therefore, we recommend the following:

1. Electron microscopy should be used to check for inclusions that are too small to identify optically.

2. If inclusions are suspected, small scale analyses of pre-selected clean garnet regions can be used to determine the background U, Th, and Pb contents and Th/U to determine meaningful cut-offs.

3. If the garnet is particularly low in U, Th, and Pb, the co-ablation of small inclusions can produce slight shifts in these elements. In this case, trace elements, such as Zr, Hf, Nb, Ce, and others by LASS-ICP-MS allows for a more sensitive cut-off values and a more thorough screening of potential contamination.

4. If trace elements cannot be collected simultaneously and inclusions are observed petrographically, the most conservative and safest approach is to reject all unstable analyses (e.g., minor undulations, positive or negative slope in the time-resolved

U, Th or Pb count rates) as potentially contaminated (in combination with U, Th, Pb, and Th/U cut-off values).

On the other hand, it may be possible to retrieve meaningful bulk U-Pb dates for micro-zircon rich garnet; however, these dates may reflect processes that pre-date garnet growth. To this point, if the 94–89 Ma U-Pb zircon micro-inclusion in garnet dates determined here do have geologic meaning, we suggest that they could reflect a low grade metamorphic or hydrothermal event during flooding of and sediment deposition or burial on the Arabian margin prior to HP-LT metamorphism.

**Data availability**

All data are included in the text and supplementary files.

**Author contributions**

JBW conceptualized the study, collected the samples, conducted petrologic and isotopic characterisation and analysis, and wrote the manuscript. All co-authors contributed to the writing of the manuscript. JMG provided additional samples for the

study, assisted with interpretation, and provided expertise on Oman geology. AB, LM, and AG assisted with setting up the



protocol and conducting LASS-ICP-MS analyses, and data processing and quality control. TG assisted with analyses and securing funding. HRM assisted in the study design, sample collection, and data interpretation.

**Competing interests**

The authors have no competing interests to declare.

**Financial support**

JBW acknowledges support from the German Science Foundation (DFG) grant 464606040. JMG acknowledges support from National Science Foundation (USA) grant EAR-2120931, as well as The Pennsylvania State University for its support of the LionChron facility. TB acknowledges support from the Johanna Quandt Young Academy Foundation, which covered costs for LASS-ICPMS analyses.

**Acknowledgements**

Andreas Scharf at Sultan Qaboos University, Oman, is thanked for technical assistance during fieldwork. We are also indebted to Silke Voigt who co-lead the excursion to Oman in February 2020.

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
