# Peer review of "Zircon micro-inclusions as an obstacle for in situ garnet U-Pb geochronology: An example from the As Sifah eclogite locality, Oman"

_EGUsphere, 2025_

## Referee Comment (RC2)

[referee-annotated manuscript omitted]

---

## Author Response (AR1)

Dear Editorial Board of Geochronology,

Attached with this letter is our revised manuscript, titled "Zircon micro-inclusions as an obstacle for in situ garnet U-Pb geochronology: An example from the As Sifah eclogite locality, Oman," for your consideration in Geochronology. The manuscript has been revised based on the reviewers' comments and the suggestions of Associate Editor Prof. Ickert. The comments were all very minor and no large structural changes have been made to the manuscript or our interpretations. We found the suggestions helpful, and we have followed nearly all the recommendations in our revision. Our responses to the reviewers' comments and edits to the manuscript are the same as those publically available in our responses in the online discussion. In the following page, we provide our feedback to the comments from Prof. Ickert.

Thank you for your time and the opportunity to revise and improve our manuscript.

Sincerely,

Jesse B. Walters, Ph.D.
Assistant Professor
NAWI Graz Geocenter, University of Graz, Graz, Austria
www.geojesse.weebly.com

**Below is our response (in italics) to the comments provided by Associate Editor Prof. Ryan Ickert:**

I would like to thank the authors and reviewers Dr. Clare Warren and Dr. Christopher McFarlane for a productive discussion on the submitted manuscript. This is a careful piece of work that demonstrates that under some circumstances, analyses of garnet U-Pb systematics by microbeam techniques can be dominated by zircon inclusions in ways that can be difficult to detect. Furthermore, they apply this observation to reconcile disparate geochronological observations in an important metamorphic locale.

Additional private note (visible to authors and reviewers only):
Dear Authors and reviewers,

Thank you for your attention to this manuscript. I recommend the authors submit a revised manuscript that address all of the reviewers points in the manner in which they describe in the replies. I have a handful of minor additional editorial comments (some of them may duplicate the reviewers comments, in which case disregard mine).

L40: Decay constants plural.

*Fixed.*

L65: Inconsistent italicization

*Fixed. No italics are used for in situ throughout.*

L70: I wouldn't say "classic". It's not obvious what that communicates.

*Removed.*

Separately, you might consider reporting the sample mass that they used (20-150 mg) to highlight the difference between this technique and a laser technique (<100 μg?). Actual numbers are useful for non-specialists.

*Added.*

L80 (approximately): I appreciate this probably came out during the final prep stages of this manuscript, but I think it would be useful to discuss the Beno et al. workflow. https://doi.org/10.1111/ggr.12561

*We are unsure what exactly about the workflow to mention here, as Beno et al. did not conduct geochronology by LA-ICPMS. However, we mention that micromilling allows for a smaller sample size than previous ID-TIMS U-Pb garnet studies.*

L247: If a laser pulse impacts the sample surface, the surface is ablated. So the four laser pulses

are not "pre-ablation". I appreciate what you are trying to say but it is inaccurate. Perhaps "…following four laser pulses used to prepare(or clean?) the sample surface"

*Fixed.*

L253: Was Pb-204 not measured?

*This is already addressed in the modified text and the rebuttal to Macfarlane's review.*

L254: An amplifier is not a "1013 ohm amplifier", it is an amplifier that uses a 1013 ohm resistor.

*Fixed.*

L255: UO and U are ions so should have a superscripted plus sign (UO+/U+). (I cannot superscript in the copernicus editorial system)

*Fixed.*

L255: What is "Th/U of ~0.9"? Is that a measured 232Th+/238U+? Or is it the interelement relative bias, suggesting Th has a 10% lower transmission (s.l.)?

*0.909 is the elemental Th/U ratio (See GEOREM data base) and this is what we are aiming for when we are tuning. The text has been modified for clarity.*

L256: The way this reference is used makes it sound like Jochum et al. made the glasses, instead of being responsible for the reference values that you use. You might consider rewriting the sentence to make attribution clearer, and/or perhaps citing a more appropriate reference, such as the summary by Kane (1998) in GGR or even the original Corning Glass publication.

*Fixed.*

L273: please consistently use the Greek symbol for sigma instead of the letter "s"

*We use s instead of sigma following the recommendation from the International Association of Geoanalysts, see: https://www.geoanalyst.org/sigma-is-out/*

L269-270: I think the authors may mean "excess scatter" and "excess variance" rather than "excess of scatter" and "excess of variance". Furthermore it is unclear whether excess scatter and excess variance are referring to the same thing, here. I assume they are? I recommend using only the term "excess scatter" because scatter is a general term, whereas "variance" has a specific statistical definition. The scatter may not be a "variance" (and indeed it's very hard to tell if it is).

*We followed Horstwood et al. 2016, which distinguish, between excess scatter derived from the scatter of the primary reference material, here NIST glass, during the sessions and the excess variance derived from the offset reference material, here the garnet reference material. There*

*was some confusion as in their Figure 1, point 5, they refer to "excess scatter" whereas in the text they refer to the "excess variance (scatter)" propagated from analyses of the primary reference materials. We have rewritten the text to make it clear that one source of variance is the reproducibility of the primary reference material, whereas another is the reproducibility of the garnet reference material.*

L271-272: The way the explanation here is written seems a bit confusing, which is unfortunate because it does sound like there has been a lot of care taken in the way the data has been reduced and reported. For example, both "within-run precision" and "counting statistics" are separated, however, within-run precision is typically calculated using the standard error of a series of sequential signal intensities (or ratios of signal intensities, or interpolated intensities/ratios), which for a well-functioning mass spectrometer, will include and possibly be dominated by counting statistics. These are not independent, and in fact are often identical. I am also confused as to how the "quoted age uncertainties" referred to on line 269 relate to "ratio uncertainties" on line 272. I don't imagine there are any great errors in this section, but it needs to be more well organized. I'd recommend just describing how the uncertainties that you use later in the manuscript are constructed from the lowest to the highest level.

*We deleted counting statistics and only mention within run precision.*

*We have clarified that the systematic "ratio" uncertainty refers to how well we know the 206Pb/238U ratio of the offset reference material (Mali).*

L293: These will be systematic at the scale of individual samples, so these are not appropriate uncertainties to use to characterize within- or between-sample variability at the scale of this study. Are these uncertainty terms large enough to matter? This may be misleading if so.

*That would be true if all analyses conducted used the same primary reference material. The reason for using NIST-SRM614 and BIR-1G for the split stream analyses were the lower contents of U, Th, and Pb, which would overwhelm the detectors on the Neptune Plus. However, this was not the case for the session using 33 um diameter spots where we measured garnet 'background' trace element contents. In this case we used GSD-1G. A full quantitative comparison between both datasets requires the propagation of the uncertainties on the reference material values, although these are relatively small.*

L294: I'm a bit confused about the statement that NIST-SRM614 are all within uncertainty of GeoREM preferred values, when NIST-SRM614 is the primary reference material for most elements (Line 287). As-written, this sounds circular, but I assume I'm missing something? Please clarify.

*This sentence should have been fixed and was a hold over from a previous version of the paper. The reason that the U content in NIST-SRM614 was off was because we used BIR-1G initially as the primary reference material for U. Because BIR-1G has so little U this resulted in a systematic offset on the U content of NIST-SRM614 when used as a secondary reference material. This is explained in the text and the sentence is modified.*

L295: I can't imagine that this makes a difference, but since there seems to be a big U discrepancy, did you use 235U at all when measuring the NIST glass? I appreciate it says 238U on line 282 but I'd like you to make sure. These glasses use DU, not natural U, and therefore have unusually high 238/235 (~418, rather than 137.818), and this can affect concentration measurements if 235U is used in any calculations. Sometimes this happens when the 238U beam is too high, for example.

*See the above response. This sentence was from an earlier version of the paper and we tracked down why there was a discrepancy. Unfortunately I missed revising this sentence.*

L296: Please be specific about which "present day Pb composition". Present day Pb from say, allanite, or low Th zircon, will have very different isotopic compositions than the average upper continental crust, for example. Simply being explicit about the IC you use will suffice.

*Fixed.*

L302: See note for L247, this is still "ablation".

*Fixed.*

L310: The run table includes 206Pb in this case. Is it not used at all? Seems like it would be a good QC check, as Pbc should have 208/206 ~ 2.0-2.2.

*LADR calculates the total common Pb from the 206Pb and 208Pb signals, not the content of 206Pb and 208Pb separately. As a result, the output of the two results is nearly 1, not 2.0-2.2. Since 208Pb is the higher signal, this value is provided.*

L318: This is a nice way to describe what you otherwise refer to as "pre-ablation".

Figure 4 caption: I think there is space to define the mineral abbreviation, there aren't very many. If you feel like you should provide a way in which a reader can interpret the abbreviations, you should make it as easy as possible.

*Fixed.*

L483: Here you say U is 0 = ng/g and on line 477 you say it is 0 µg/g. Of course, they are equivalent, but it is confusing to mix the SI-prefixes in this way.

*Fixed.*

It is also peculiar to state that they are exactly zero, surely the concentration is just some amount that is below detection limits? Please clarify.

*Those are the assumed upper and lower limits of the mixing lines. Whether the lower intercept is at U = 0 µg/g or 0.001 µg/g will have no effect on the plotted mixing lines. The data clearly converge roughly to a y intercept of 0. Again, these mixing lines are for visual comparison and*

*do not represent fits to the data.*

L538: Please do not use the redundant phrase "age dating".
*Fixed*

L545: You have switched from % above the paragraph to ‰ here, used only once. That is confusing to a reader and unnecessary: 0.2% is still very legible. Please change this to %.

*Fixed.*

L596: As above, the use of ‰ is unnecessary, please change to %.

*Fixed.*

L628: Please add superscripts.

*Fixed.*

L637: It's not necessary to specify cps, you can eliminate the parenthetical.

*Fixed.*

L700: This sentence before the comma is incomplete, or it otherwise does not make sense.

*Fixed.*

L726: Pretty sure most Lu-Hf is by MC-ICPMS, not TIMS.

*Correct, fixed.*